# Small molecules targeting the disordered transactivation domain of the androgen receptor induce the formation of collapsed helical states

Jiaqi Zhu[1], Xavier Salvatella [2,3] & Paul Robustelli [1] ✉

Intrinsically disordered proteins, which do not adopt well-defined structures under physiological conditions, are implicated in many human diseases. Small molecules that target the disordered transactivation domain of the androgen receptor have entered human trials for the treatment of castration-resistant prostate cancer (CRPC), but no structural or mechanistic rationale exists to explain their inhibition mechanisms or relative potencies. Here, we utilize all-atom molecular dynamics computer simulations to elucidate atomically detailed binding mechanisms of the compounds EPI-002 and EPI-7170 to the androgen receptor. Our simulations reveal that both compounds bind at the interface of two transiently helical regions and induce the formation of partially folded collapsed helical states. We find that EPI-7170 binds androgen receptor more tightly than EPI-002 and we identify a network of intermolecular interactions that drives higher affinity binding. Our results suggest strategies for developing more potent androgen receptor inhibitors and general strategies for disordered protein drug design.

Intrinsically disordered proteins (IDPs) that lack a fixed three-dimensional structure under physiological conditions represent ~40% of the human proteome, have crucial functional roles in a variety of biological pathways and biomolecular assemblies, and are implicated in many human diseases[1–8]. IDPs represent an enormous pool of potential drug targets that are currently inaccessible to conventional structure-based drug-design methods utilized for folded proteins[9–15]. In conventional drug-design campaigns, small molecules are designed to optimize the strength of intermolecular interactions with well-defined binding sites. The structures of drug binding sites in folded proteins are often known in their apo and bound forms from high-resolution crystal structures. In cases where bound structures of inhibitors are known, it is often straightforward to identify pharmacophores to optimize with small molecule modifications. A number of computational tools exist to predict the affinities of small molecules

that bind structured binding sites in folded proteins, with alchemical free-energy perturbations calculations growing in popularity and offering increasingly accurate predictions relative to more high-throughput docking and virtual screening approaches[16,17].

Unlike folded proteins, IDPs populate a conformational ensemble of rapidly interconverting structures in solution[18]. The conformations of IDPs can sample a large number of topological arrangements, and structures in IDP conformational ensembles can have little-to-no structural similarity to one another. Many IDPs possess sequence elements with elevated secondary structure populations relative to random-coil distributions[18,19]. This confers a limited amount of local order to IDP conformational ensembles, but the number of thermally accessible orientations of sidechain and backbone pharmacophores can remain combinatorially large. As such, it is generally not possible to represent even small sequence segments of IDPs by a single

[1]Dartmouth College, Department of Chemistry, Hanover, NH 03755, USA. [2]Institute for Research in Biomedicine (IRB Barcelona), The Barcelona Institute of Science and Technology, Baldiri Reixac 10, 08028 Barcelona, Spain. [3]ICREA, Passeig Lluís Companys 23, 0810 Barcelona, Spain. ✉e-mail: Paul.J.Robustelli@Dartmouth.edu

dominant conformation or a small number of substantially populated conformations. IDPs are, therefore, not suitable targets for conventional structure-based drug-design methods that require the existence of an ordered binding site, and the general "druggability" of IDP sequences remains uncertain[13].

If it becomes possible to rationally target IDPs with small molecule drugs, the druggable proteome will be dramatically expanded and therapeutic interventions may become accessible for currently untreatable diseases associated with aberrant biological interactions of IDPs such as those mediated by biomolecular condensate formation and protein misfolding[10,15,20–23]. Several small molecules have been discovered that interact with specific IDP sequences and inhibit their interactions[24–31], and many of these compounds show clear biological activity[24–27]. Biophysical measurements, particularly small but reproducible NMR chemical shift perturbations observed in ligand titrations, indicate that these IDPs remain highly dynamic while interacting with these compounds[24–26,30,31]. These observations suggest that the sequence specificity of these ligands is conferred through a network of transient interactions and that these bound states consist of heterogeneous ensembles of interconverting binding modes.

A number of all-atom molecular dynamics (MD) simulation studies provide support for dynamic binding mechanisms between IDPs and small molecule ligands[30,32–36]. In several studies, ligands are observed to populate a broad distribution of binding modes that confer little-to-no detectable ordering in IDP binding sites relative to their apo forms[32–34]. In a recent joint MD and NMR study a series of small molecule ligands were found to bind specifically to the C-terminal region of α-synuclein with differing affinities without significantly altering the conformational ensembles of the apo and bound states[32]. This study proposed that the specificity and affinity of these ligands is conferred through a so-called dynamic shuttling mechanism where a ligand rarely forms multiple specific intermolecular interactions simultaneously and instead transitions among networks of spatially proximal interactions. In this mechanism, differences in the affinity and specificity of ligands are attributed to the complementarity of the orientations of protein and ligand pharmacophores in a dynamic IDP ensemble without evoking the notion of an ordered binding site. In some cases, biophysical experiments and molecular simulation data suggest that the conformational ensembles of IDPs undergo an entropic expansion upon ligand binding, where interactions with a small molecule can increase the number of conformations significantly populated by an IDP[27,28,36]. In other cases, biophysical experiments and simulation results suggest that small molecule ligands can cause a population shift among existing IDP conformations[29], drive the compaction of monomeric disordered state proteins upon binding[30,35] or drive the formation of soluble oligomeric states[31]. The variety of binding mechanisms observed suggests that small molecules affect the conformational ensembles of IDPs in a system-dependent manner[10,11].

While a number of small molecules have been found to bind and inhibit IDPs in vitro and in vivo, one of the most pharmaceutically promising IDP:drug interactions is the inhibition of the disordered N-terminal transactivation domain (NTD) of the androgen receptor (AR) by a series of compounds with a common Bisphenol-A scaffold[37–42]. AR is a large multidomain transcription factor that contains folded ligand binding and DNA-binding domains in addition to the 558-residue intrinsically disordered NTD[24]. Transcriptionally active AR drives the growth of most prostate cancers[14]. In ~65% of cases, prostate cancer patients can be cured by surgery or radiotherapy. In the remaining ~35% of patients, the progression of the disease can initially be treated with a number of FDA-approved drugs that target the folded AR ligand binding domain by competitively inhibiting the binding site of transcription-activating androgens. After a period of 2–3 years, however, these patients inevitably become refractory to pharmacological interventions and develop lethal castration-resistant prostate cancer (CRPC). In CRPC, cancer cells acquire mutations and

express splice variants that enable the activation of AR at low levels of circulating androgens and in the presence of ligand-binding domain inhibitors. One of the most common CRPC resistance adaptions is the expression of constitutively active AR splice variants that lack the ligand-binding domain entirely[43], and therefore render all current FDA-approved drugs ineffective.

The discovery of small molecules that target the disordered AR-NTD provides a promising therapeutic strategy for CRPC, as a number of these compounds have been shown to inhibit the constitutively active splice variants of AR that lack the ligand-binding domain[14]. A series of AR-NTD inhibitors were discovered from screening libraries of natural products and were shown to have efficacy in CRPC mouse models[37–42]. Among these compounds, EPI-002, a bisphenol-A derivative later named Ralaniten, was identified as a promising AR-NTD inhibitor and a prodrug of EPI-002 (EPI-506; Ralaniten Acetate) was selected for phase I clinical trials in 2015 (https://clinicaltrials.gov/ct2/show/NCT02606123), becoming the first small molecule known to directly bind an intrinsically disordered protein to be tested in humans[14]. Human trials were, however, discontinued after phase I due to insufficient potency and poor metabolic properties. The compound EPI-7170, a second-generation AR-NTD inhibitor, was subsequently shown to have dramatically improved potency and metabolic properties relative to EPI-002[40–42], and an additional second-generation AR-NTD inhibitor with an undisclosed chemical structure (EPI-7386) entered phase I clinical trials in March 2020 (https://clinicaltrials.gov/ct2/show/NCT04421222)[44].

Nuclear magnetic resonance (NMR) spectroscopy has localized the strongest interactions of EPI-002 to the transactivation unit 5 domain (Tau-5; residues A350-C448) of the AR-NTD[24]. The Tau-5 domain contains three regions with transiently populated helices termed R1, R2, and R3[24,45,46]. These domains were found to have the largest NMR chemical shift perturbations (CSPs) in EPI-002 titrations[24]. The atomic details of the molecular mechanisms by which EPI-002 and other EPI compounds interact with the Tau-5 are, however, not well understood and no structural or mechanistic rationale exists to explain their affinity to the AR-NTD or their relative efficacies in CRPC treatment. In the absence of such mechanistic understanding, it is not currently possible to rationally design new compounds with improved potencies or to utilize similar inhibition mechanisms to target other intrinsically disordered domains.

In this investigation, we utilize all-atom explicit solvent MD simulations with the state-of-the-art a99SB-disp force field[47] to study the binding mechanisms of EPI-002 and EPI-7170 to the transiently helical Tau-5 region of the disordered AR-NTD. We observe that EPI-002 and EPI-7170 both induce the formation of compact helical molten-globule-like states in Tau-5. These bound states remain dynamic and sample a heterogeneous ensemble of binding modes. We find that EPI-7170 has a 2.5-fold higher affinity to Tau-5 than EPI-002 and that the EPI-7170 bound ensemble is substantially more helical relative to the bound ensemble of EPI-002. We identify a network of interactions in the EPI-7170 bound ensemble that more effectively stabilize these collapsed helical conformations. Our results suggest that EPI compounds inhibit the activity of AR by inducing the partial folding of molecular recognition elements in the Tau-5 domain into compact helical states and preventing interactions between AR and the transcriptional machinery required for elevated AR transactivation in CRPC patients.

EPI-002 and EPI-7170 both possess a chlorohydrin group that has been shown to be weakly reactive with Tau-5, and it is hypothesized that covalent attachment of EPI compounds to Tau-5 may be required for its biological activity[14,37]. Non-covalent binding may therefore be the first step in AR inhibition which localizes reactive ligands to specific cysteines in the AR-NTD and increases their rate of covalent attachment. The atomically detailed binding mechanisms described here reveal that higher affinity non-covalent binding of EPI-7170 to Tau-5

substantially increases the proximity of the EPI-7170 chlorohydrin group to the reactive thiol of residue cysteine 404 relative to the proximity of the EPI-002 chlorohydrin and cysteine 404 thiol groups observed in simulations of EPI-002 binding. We hypothesize that the non-covalent binding of EPI-7170 and EPI-002 described here increases the local concentration of these compounds near weakly reactive cysteines in Tau-5, driving covalent attachment and potentially further stabilizing the formation of transcriptionally inactive compact helical states. Our results suggest an inhibition mechanism where the site-specific covalent reactivity of weak nucleophilic groups in IDP ligands is dictated by the non-covalent affinity of IDP ligands for specific binding sites. The atomically detailed binding mechanisms described here suggest strategies for developing more potent AR inhibitors for the treatment of CRPC, and general strategies for targeting intrinsically disordered drug targets.

## Results

### Molecular dynamics simulations of the disordered Tau-5$_{R2\_R3}$ region of the androgen receptor N-terminal transactivation domain are consistent with NMR experiments

Previously reported NMR chemical shift perturbations (CSPs) indicate that EPI-002 interacts non-covalently with a subset of three partially helical regions, termed R1, R2, and R3, in the transactivation unit 5 (Tau-5) domain (residues 350–448) of the disordered AR-NTD[24]. Circular Dichroism experiments and NMR secondary chemical shifts suggest that R1, R2, and R3 are -10, 30, and 10% helical, respectively, when Tau-5 is in its apo form in solution[24,45,46,48]. The 31-residue R1 region (residues S341-G371) is separated from the 24-residue R2 region (residues L391-G414) by a 19-residue proline-rich linker region that appears to predominantly populate an extended polyproline II conformation based on sequence composition and secondary NMR chemical shifts[24,48]. The R2 region is separated from the R3 region (residues S426-G446) by a glycine-rich linker ($^{415}$AGAAGPGSGSP$^{425}$) that is predicted to have no residual secondary structure propensity based on sequence and secondary chemical shifts (Fig. 1)[24,48].

The relatively small magnitude of the backbone amide CSPs measured in Tau-5 upon EPI-002 binding suggests that this interaction does not induce rigid folded structures in these regions, and that the Tau-5 region remains highly dynamic while interacting with this compound[24]. Experiments were also performed to determine if the titration of peptides containing isolated R1, R2, and R3 domains produced CSPs or line-broadening effects on the NMR resonances of EPI-

001 (a racemic mixture containing EPI-002 along with three additional stereoisomers)[24]. No interactions were detected between EPI-001 and these truncated partially helical constructs, suggesting that multiple partially helical regions are required to detect the interaction between EPI-002 and Tau-5 by NMR. These data suggest that EPI-002 may be binding at the interfaces of these transiently helical domains and stabilizing long-range tertiary contacts found in the apo Tau-5 ensemble.

The largest backbone amide NMR CSPs observed in EPI-002:Tau-5 titrations are found in the R2 and R3 regions. Specifically, the largest individual CSPs were found in the most helical residues of R2 ($^{400}$AAQ$^{403}$) and the $^{433}$WHTLF$^{437}$ segment of R3. The $^{433}$WHTLF$^{437}$ segment has been found to be critical for the function AR and to fold into a helical conformation upon binding the RAP74 domain of the general transcription regulator TFIIF[45,46,49,50]. The disruption of this interaction causes AR to lose its transcriptional activity[37,38,50]. Based on these observations and the sequence composition of Tau-5, we decided to focus our investigation on a 56-residue Tau-5 fragment (residues L391-G446) containing the R2 and R3 helices, which we refer to as Tau-5$_{R2\_R3}$. Given the extended nature of the proline-rich linker region between R1 and R2, and the small magnitude of CSPs in the R1 region upon EPI-002 binding[24], we hypothesize that transient long-range contacts between R1 and R2 and R1 and R3 are less likely to be important for EPI-002 and EPI-7170 binding than shorter range contacts between R2 and R3. Simulating the smaller Tau-5$_{R2\_R3}$ fragment enables us to focus our investigation on the region of Tau-5 that showed the most pronounced CSPs in EPI-002 titrations and obtain better statistics on a more tractable sampling problem.

We ran an unbiased all-atom explicit solvent MD simulation of Tau-5$_{R2\_R3}$ with the a99SB-*disp* protein force field and a99SB-*disp* water model[47] using the replica exchange with solute tempering (REST2) enhanced sampling algorithm[51] (See Methods). Simulations with the a99SB-*disp* force field have been found to provide excellent agreement with the secondary structure propensities and the dimensions of IDPs[47]. The apo Tau-5$_{R2\_R3}$ REST2 MD simulation was run with 16 replicas, spanning temperatures from 300–500 K, with all protein atoms selected for solute tempering. The simulation was run for 4.6 µs/replica, for an aggregate simulation time of 74 µs. The convergence of the simulation was assessed by computing statistical error estimates by a blocking analysis[52,53] (See Methods) and comparing secondary structure propensities, intramolecular contact maps, and free energy surfaces as a function of the radius of gyration ($R_g$) and the α−helical order parameter $S\alpha$[54] for each temperature rung in the

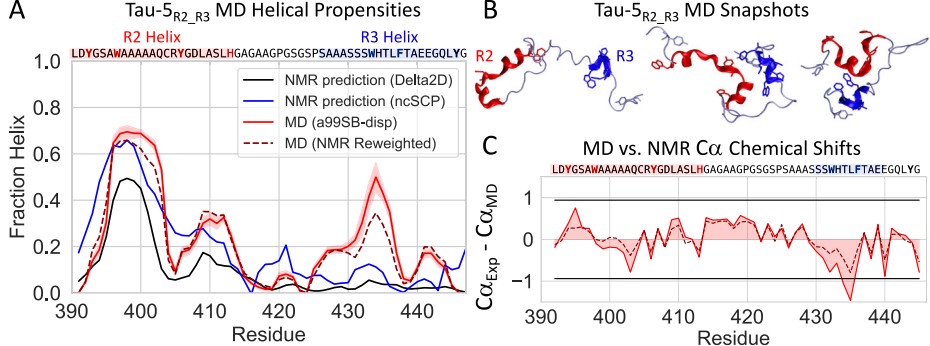

**Fig. 1 | Molecular dynamics simulations of the disordered Tau-5$_{R2\_R3}$ region of the androgen receptor N-terminal transactivation domain. A** Comparison of the helical propensity observed in the 300 K replica of a 74 µs explicit solvent REST2 MD simulation of Tau-5$_{R2\_R3}$ in its apo form (red), the helical propensity obtained after maximum-entropy reweighting of the MD ensemble using NMR Cα chemical shifts as restraints (discontinuous line, dark red) and helical propensity predictions derived from experimental NMR chemical shifts (blue and black). Helical propensities were calculated with the DSSP algorithm. Simulated helical propensities are presented as mean values ± statistical error estimates from blocking. **B** Illustrative snapshots from an MD simulation of the Tau-5$_{R2\_R3}$. The R2 and R3 regions of Tau-5$_{R2\_R3}$ are colored red and blue, respectively. **C** Comparison of experimental NMR Cα chemical shifts with shifts calculated from MD simulations using SPARTA+ before (red) and after maximum-entropy reweighting (discontinuous line, dark red). Black lines indicate the average Cα chemical shift prediction error of SPARTA+ on its training database of folded protein structures. Source data are provided as a Source Data file.

**Table 1 | Agreement between calculated and experimental NMR chemical shifts from the 300 K replica of a 74 µs unbiased REST2 MD simulation of Tau-5$_{R2\_R3}$ employing the a99SB-*disp* force field and from a maximum-entropy reweighted ensemble derived using Cα NMR chemical shifts as restraints**

| | Cα | HN | N | C' | Cβ |
|---|---|---|---|---|---|
| Unbiased Tau-5$_{R2\_R3}$ MD Ensemble | | | | | |
| RMSD (ppm) | 0.47 | 0.23 | 1.06 | 0.45 | 0.37 |
| Correlation | 0.991 | −0.558 | 0.954 | 0.915 | 1.000 |
| Cα-reweighted Maximum Entropy Tau-5$_{R2\_R3}$ Ensemble | | | | | |
| RMSD (ppm) | 0.29 | 0.21 | 0.88 | 0.38 | 0.33 |
| Correlation | 0.997 | −0.312 | 0.968 | 0.934 | 1.000 |

Chemical shifts were calculated using SPARTA+[55].

REST2 simulation and for each independent demultiplexed replica (See Methods, Figs. S1–S6, Supporting Information: "Convergence Analyses").

The average helical propensity observed in the REST2 MD simulation of apo Tau-5$_{R2\_R3}$ and illustrative snapshots from the 300 K replica are shown in Fig. 1. Additional representative snapshots are shown in Fig. S7. The intramolecular contact map of Tau-5$_{R2\_R3}$ reveals highly populated (~60–80% occupancy) contacts between the partially helical $^{396}$AWAAAAAQCRY$^{406}$ residues in R2 and the $^{432}$SWHTLF$^{437}$ residues in R3 (Fig. S3). Statistical error estimates of the simulated properties of the apo Tau-5$_{R2\_R3}$ ensemble simulation were calculated using a blocking analysis following refs. 52,53 (See Methods). Using this approach, we calculated error estimates for secondary structure propensities (Average helical content = 23.3 ± 0.6%), radius of gyration ($R_g$ = 12.5 ± 0.1 Å) and the helical order parameter Sα[54] (Sα = 6.5 ± 0.1). Sα reports on the similarity of each six-residue fragment in the protein to an ideal helical conformation. Each six-residue fragment with a small RMSD (<0.5 Å) from an ideal helical conformation contributes a value of ~1 to the total Sα value, while six-residue fragments with a large RMSD (>3.0 Å) contribute a value of ~0 to the total Sα value (See Methods). The value of *S*α for a protein structure can therefore be interpreted as a proxy for the number of six-residue fragments closely resembling an ideal helical conformation. Together our convergence analyses (Figs. S1–S6) and statistical error estimates suggest that our apo Tau-5$_{R2\_R3}$ REST2 MD simulation is well converged. A subset of conformations observed in the 300 K replica of our apo Tau-5$_{R2\_R3}$ MD simulation are shown in Movie S1.

We validated the accuracy of our MD simulation of Tau-5$_{R2\_R3}$ by calculating NMR chemical shifts with SPARTA+[55] and comparing them to previously reported experimental values[24] (Table 1 and Fig. 1). The overall agreement between calculated and experimental backbone chemical shifts is excellent and is on-par with the most accurate chemical shift predictions reported in force field benchmarks of simulations of IDPs[47,56]. Specifically, we note that the RMSD between calculated and experimental NMR chemical shifts observed for this simulation (Cα RMSD = 0.47 ppm) is among the lowest values reported for unbiased MD simulations of disordered proteins in an extensive benchmark long tine-scale MD simulations of nine IDPs run with seven state-of-the-art force fields, and is lower than the RMSD between calculated experimental shifts for several experimentally derived IDP ensembles obtained using NMR chemical shifts as restraints[47]. The predicted Cα shifts are within the estimated 0.92 ppm SPARTA + Cα shift prediction error for all residues with the exception of slightly larger deviations in residues $^{434}$WH$^{435}$ (Fig. 1C). In addition to a direct comparison with NMR chemical shifts, we also observed that the simulated helical propensity of residues $^{431}$SSWHTLF$^{437}$ in the R3 region, calculated using the DSSP algorithm, are somewhat overestimated relative to helical propensities estimated from NMR

chemical shifts using the secondary structure propensity prediction algorithms Delta2D and ncSCP[57,58] (Fig. 1A). We note that chemical shift-based secondary structure propensity prediction algorithms such as Delta2D and ncSCP are subject to systematic and sequence-specific errors and show larger deviations with helical propensity estimates from circular dichroism (CD) for secondary structure elements with a population <30%[57].

To rigorously quantify the error in the simulated helical propensity directly against experimental data, we utilized the maximum-entropy reweighting algorithm of refs. 59,60 (See Methods) to reweight our unbiased 300 K ensemble using Cα chemical shifts as restraints (Table 1 and Fig. 1). We note that while only Cα chemical shift predictions were used as restraints in the reweighting procedure, we observed improvements in the prediction accuracy of the remaining backbone shifts in the reweighted ensemble, suggesting that the resulting ensemble is not overfit to the Cα chemical shift data. We found that optimal agreement with experimental shifts was obtained by reducing the average helical propensity of residues $^{431}$SSWHTLF$^{437}$ calculated using the DSSP algorithm from 34% in the unbiased ensemble to 25% in a reweighted ensemble, suggesting an over-stabilization of helical conformations in this region by 0.26 kcal/mol (or -0.04 kcal/mol per residue in this seven-residue segment) in the unbiased trajectory. This suggests that improved agreement with NMR chemical shifts could potentially be obtained in unbiased simulations by applying a residue-specific force field torsion correction to these residues[59] however, free energy differences this small may be difficult to resolve beyond the statistical uncertainty of conformational sampling in simulations of IDPs of this size.

We note that the Cα-reweighted maximum entropy ensemble still possesses more helical content in the R3 region than is predicted by the NMR chemical shift-based algorithms Delta2D and ncSCP, though this level of discrepancy is potentially within the error of these secondary structure propensity prediction methods (Fig. 1A). Considering that both chemical shift predictions from SPARTA+[55] and NMR chemical shift based secondary structure prediction algorithms[55,57,58] are subject to prediction errors, we consider the relatively modest deviation from experimental chemical shifts and predicted secondary structure propensities to be acceptable.

We note that while unbiased Tau-5$_{R2\_R3}$ simulations run with the a99SB-*disp* force field may slightly overestimate the stability of the R3 helix, the simulated helical conformations in these regions are still only marginally stable with an average population of 34% in the 300 K replica of our REST2 simulation. As these simulations sample both helical and non-helical states, we expect that unbiased simulations will appreciably sample ligand binding modes with both helical and non-helical states of the R3 region. We, therefore, expect that an unbiased simulation should be capable of resolving increases or decreases in the stability of helical conformations in the presence of ligands, even if the absolute populations of helical conformations in simulated bound and unbound states are elevated relative to solution experiments. It is possible, however, that a slight overstabilization of helical conformations in the R3 region in the apo ensemble of Tau-5$_{R2\_R3}$ may make an increase in the stability of helical conformations in this region in the presence of ligands more difficult to detect, and simulations in the presence of ligands may underestimate increases in the helical propensity of the R3 region in the presence of ligands.

## EPI-7170 has a higher affinity to Tau-5$_{R2\_R3}$ than EPI-002

We utilized the same REST2 MD simulation protocol used for apo simulations of Tau-5$_{R2\_R3}$ to perform unbiased simulations of Tau-5$_{R2\_R3}$ in the presence of EPI-002 and EPI-7170 (See Methods). EPI-002 and EPI-7170 were parameterized with the GAFF1 forcefield[61]. Simulations run with GAFF1 and a99SB-*disp* force fields have previously been shown to provide excellent agreement with residue-specific IDP ligand binding propensities based on comparisons to NMR CSPs[30,32]. A

**Table 2 | Simulated values and statistical error estimates for properties of Tau-5$_{R2\_R3}$ in its apo form and when bound to EPI-002 and EPI-7170**

| | R$_g$ (nm) | Fraction helix | Sα | Bound fraction | Ligand $K_D$ (mM) | Helical globule population | ΔG$_{globule}$ (kcal/mol) |
|---|---|---|---|---|---|---|---|
| Apo Tau-5$_{R2\_R3}$ | 1.25 ± 0.01 | 23.3 ± 0.6% | 6.5 ± 0.1 | - | - | 40.4 ± 5.2% | +0.23 ± 0.13 |
| EPI-002:Tau-5$_{R2\_R3}$ Bound Ensemble | 1.26 ± 0.3 | 25.4 ± 1.3% | 7.3 ± 0.4 | 43 ± 2% | 5.24 ± 0.43 | 48.5 ± 6.5% | +0.04 ± 0.17 |
| EPI-7170:Tau-5$_{R2\_R3}$ Bound Ensemble | 1.24 ± 0.02 | 32.8 ± 0.5% | 9.6 ± 0.2 | 67 ± 2% | 1.92 ± 0.15 | 61.1 ± 4.5% | −0.27 ± 0.11 |

Sα is an α–helical order parameter that is a proxy for the number of six-residue helical fragments present in a protein conformation. Collapsed "helical globule" states are defined as Tau-5$_{R2\_R3}$ conformations with values of Sα > 6.0 and R$_g$ < 1.3 nm. ΔG$_{globule}$ is the free energy of formation of the helical globule state at 300 K. Values for apo Tau-5$_{R2\_R3}$ were calculated from a simulation of Tau-5$_{R2\_R3}$ performed in the absence of ligands. Values for the EPI-002:Tau-5$_{R2\_R3}$ and EPI-7170:Tau-5$_{R2\_R3}$ bound ensembles were calculated using only bound frames in simulations in the presence of ligands. Error estimates were computed using a blocking analysis[52,53].

REST2 simulation of EPI-002 and Tau-5$_{R2\_R3}$ was run for 4.0 µs/replica, for an aggregate simulation time of 64 µs, and a REST2 simulation of EPI-7170 and Tau-5$_{R2\_R3}$ was run for 4.5 µs/replica, for an aggregate simulation time of 72 µs. The simulated properties of Tau-5$_{R2\_R3}$ observed in these simulations are compared to the apo Tau-5$_{R2\_R3}$ simulations in Table 2. Simulation convergence analyses are reported in Figs. S8–S19.

To calculate a simulated $K_D$ value for each compound, we define the bound population (P$_b$) of each ligand as the fraction of frames with at least one intermolecular contact between the ligand and Tau-5$_{R2\_R3}$, where intermolecular contacts are defined as occurring in frames where at least one ligand-heavy (non-hydrogen) atom is within 6.0 Å of a protein-heavy atom. This cutoff was selected to reflect the distance that non-bonded interactions have been shown to have measurable effects on the chemical shifts of protein[55,62]. We calculated the $K_D$ value according to $K_D = P_u/P_b(vc°N_A)^{-1}$ where (P$_u$) is the fraction of frames with no ligand contacts, $v$ is the volume of the simulation box, $N_A$ is Avogadro's number, $c°$ is a standard state concentration (1 mol L$^{-1}$), and $(vc°N_A)^{-1}$ is the simulated concentration[63]. The simulated concentration of 1 copy of Tau-5$_{R2\_R3}$ in the 7.5 nm box used here is 3.93 mM. Using these definitions, we observe that EPI-002 has a bound population of 43 ± 2%, corresponding to a $K_D$ value of 5.24 ± 0.43 mM and EPI-7170 has a bound population of 67 ± 2%, corresponding to a simulated $K_D$ value of 1.92 ± 0.15 mM. We report the simulated $K_D$ values and estimate statistical errors as a function of simulation time in Fig. S22, illustrating that these quantities are well converged and that the difference in simulated $K_D$ values is statistically significant. In a previous study of small molecules binding to a 20-residue fragment of α-synuclein[32] that employed a more lenient 6.0 Å contact threshold between all protein atoms (including hydrogens) and ligand-heavy atoms, the lowest simulated $K_D$ value observed among a series of 50 ligands was 4.5 mM ± 0.15 mM. The simulated affinity of EPI-002 to Tau-5$_{R2\_R3}$ is therefore on-par with the tightest binding ligands from that study, and the simulated K$_D$ of EPI-7170 to Tau-5$_{R2\_R3}$ is ~2.5-fold smaller than the tightest binding ligand from that study[32].

We note that the absolute values of intermolecular contact probabilities and the corresponding $K_D$ values will be sensitive to the distance thresholds used to define intermolecular contacts and that a purely distance-based definition of "bound" frames will also count transient collisions between ligands and proteins among bound conformations. This is not inherently problematic, however, as experimental spectroscopic methods used to detect ligand binding such as NMR spectroscopy, may also be sensitive to such transient collisions. We note that different spectroscopic methods used to measure experimental binding affinities of ligands to disordered proteins will be sensitive to different features of ligand binding modes to varying extents. In particular, atomic resolution spectroscopy, such as NMR, will detect interactions with each residue independently, and the strength of the signal will depend on the identity of the chemical moieties that are brought into proximity upon binding. Measurements from surface plasmon resonance (SPR), biolayer interferometry, isothermal calorimetry (ITC), or fluorescence anisotropy may be more globally sensitive to binding and report on the total number of molecules in solution that contain any contacts with ligands. In several instances, small molecule ligands that appear to bind IDPs with relatively weak mM affinities as assessed by residue-level NMR chemical shift perturbations appear to bind substantially tighter, in the low µM affinity range, using other spectroscopic techniques such as SPR and biolayer interferometry[27,28,30]. Understanding these relationships well enough to quantitively compare simulated and spectroscopically measured binding affinities will likely require extensive experimental and computational benchmarking. Simulated $K_D$ values are, therefore, most meaningfully compared to simulated $K_D$ values calculated with the same distance thresholds and may not be directly comparable to experimental $K_D$ values measured from biophysical experiments and spectroscopic techniques beyond ranking the relative affinities of ligands in a series.

We and others[32] have found that the distance threshold used to define intermolecular contacts has little effect on the ratios of $K_D$ values and contact probabilities calculated for a series of small molecule ligands in separate binding simulations. The distance threshold employed in the definition of bound conformations can therefore be thought of as a scaling factor for simulated $K_D$ values and the optimal distance threshold for comparisons to experimental $K_D$ values is likely to vary based on the experimental techniques used. In single replica unbiased MD simulations, one can utilize the residence time of ligand binding events to differentiate transient encounters from more stable binding events[32], but applying residence time-based thresholds for bound conformations is less straightforward when using enhanced sampling techniques such as REST2[51]. In a previous study of small molecules binding a 20-residue fragment of α-synuclein that utilized unbiased MD simulations and the same force fields employed in this investigation[32], a broad distribution of residence times, from 1 ns–2 µs was observed. Based on the higher simulated binding affinities of the compounds studied here, we speculate that the distribution of residence times observed in unbiased MD simulations of EPI-002 and EPI-7170 binding Tau-5$_{R2\_R3}$ would be shifted to longer residence times.

We further note that EPI-002, which produces small NMR CSPs consistent with an mM in vitro binding affinity and has a simulated mM $K_D$ value, has been shown to have ~10 µM IC$_{50}$ value in prostate-specific antigen luciferase reporter assays for inhibition of endogenous AR transcriptional activity in cellular assays as well as clear anti-tumor activity in mouse models[37]. We, therefore, caution that the magnitudes of experimental $K_D$ values from biophysical assays and simulated $K_D$ values from molecular dynamics computer simulations are not clear predictors of the biological activity of IDP ligands. There is no clear evidence to suggest that nanomolar affinity binding is required for IDP ligands to exhibit biological activity. This likely results from the fact that IDPs often have central roles in cellular signaling networks that can involve multivalent low-affinity interactions with a large number of

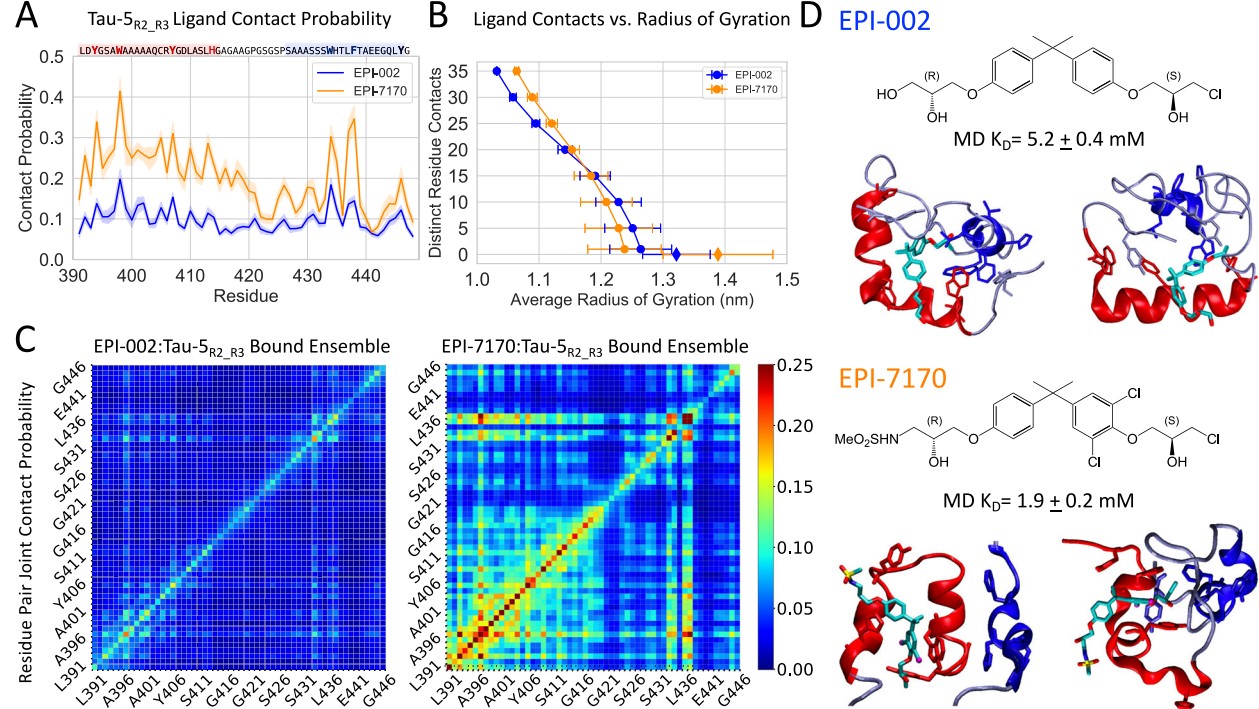

**Fig. 2 | Molecular dynamics simulations of EPI-002 and EPI-7170 binding the disordered Tau-5$_{R2\_R3}$ region of the androgen receptor N-terminal transactivation domain. A** Per-residue contact probabilities were observed between Tau-5$_{R2\_R3}$ and EPI-002 (blue) and EPI-7170 (orange) in REST2 MD simulations. Contacts are defined as occurring in frames where any non-hydrogen ligand atom is within 6.0 Å of a non-hydrogen protein atom. Contact probabilities are reported as mean values ± statistical error estimates from blocking. **B** Average Cα-atom radius of gyration (R$_g$) of Tau-5$_{R2\_R3}$ conformations as a function of the minimum number of distinct residue contacts formed in MD simulations. Values are presented as means and error bars reflect the variance of R$_g$ values observed in each subset of Tau-5$_{R2\_R3}$

conformations. Diamonds reflect the average radius of gyration of Tau-5$_{R2\_R3}$ conformations in frames with no protein-ligand contacts in simulations in the presence of ligands. **C** The probability that a pair of residues in Tau-5$_{R2\_R3}$ simultaneously form ligand contacts in the EPI-002:Tau-5$_{R2\_R3}$ and EPI-710:Tau-5$_{R2\_R3}$ bound ensembles. **D** Chemical structures of EPI-002 and EPI-7170 and illustrative MD Snapshots of Tau-5$_{R2\_R3}$ interacting with ligands. The R2 and R3 helices of Tau-5$_{R2\_R3}$ are colored red and blue, respectively. $K_D$ values observed in MD simulations were calculated by defining the bound population (P$_b$) as the fraction frames with at least one ligand contact, and the reported statistical error estimates were obtained from a blocking analysis. Source data are provided as a Source Data file.

physiological interaction partners, as well as the importance of both kinetics and thermodynamics in the formation of higher-order molecular species involved in protein misfolding and biomolecular condensate formation in cellular contexts[8,9,20–23].

## EPI-002 and EPI-7170 predominantly interact with aromatic residues at an interface between the R2 and R3 regions of Tau-5$_{R2\_R3}$

We observed that bound states of EPI-002 and EPI-7170 are highly dynamic and consist of a heterogenous ensemble of interconverting binding modes, consistent with the previously proposed dynamic shuttling mechanism[32] (Fig. 2, Figs. S20, S21, and Movies S2, S3). This observation is consistent with the relatively small magnitude of NMR CSPs observed in EPI-002 ligand titrations[24]. The contact probability of EPI-002 and EPI-7170 with each residue of Tau-5$_{R2\_R3}$ are shown in Fig. 2A. Convergence analyses of the per-residue contact probabilities are shown in Figs. S23, S24. We observe that EPI-7170 has an elevated contact probability with all residues relative to EPI-002 and that the most populated intermolecular contacts of both ligands are made with aromatic residues in the R2 and R3 regions.

The most populated contacts between EPI-002 and Tau-5$_{R2\_R3}$ are made by residues W397 and W433, and the most populated contacts between EPI-7170 and Tau-5$_{R2\_R3}$ are made by residues W397 and F437. We observed highly populated intramolecular contacts between these residue pairs in simulations of apo Tau-5$_{R2\_R3}$ (Figs. S3, S25). To determine if the EPI compounds are binding at an interface formed by the R2 and R3 regions, we examined the probability of EPI-002 and EPI-7170 forming simultaneous contacts with each pair of residues in Tau-

5$_{R2\_R3}$ (Fig. 2C). We observe substantially more cooperative binding to residues in both the R2 and R3 regions in the EPI-7170 bound ensemble. The bound ensembles of EPI-002 and EPI-7170 contain contacts with aromatic residues in both the R2 and R3 regions in 46.7 and 60.0% of bound conformations, respectively. Our simulations, therefore, indicate that both EPI-002 and EPI-7170 predominantly bind at an interface between the R2 and R3 region of Tau-5$_{R2\_R3}$, and that EPI-7170 does so to a greater extent.

## EPI-002 and EPI-7170 binding induces the formation of collapsed helical molten-globule-like states of Tau-5$_{R2\_R3}$

In both EPI-002 and EPI-7170 bound ensembles, we found that bound conformations that form simultaneous contacts with greater numbers of Tau-5$_{R2\_R3}$ residues become progressively more compact (Fig. 2B), suggesting a scenario where the R2 and R3 regions collapse around the ligands as they penetrate into the hydrophobic core of Tau-5$_{R2\_R3}$. We provide illustrative conformations of EPI-002 and EPI-7170 bound to more compact conformations Tau-5$_{R2\_R3}$ in Fig. 2D, Figs. S20, S21, and Movies S2, S3. We have visualized the free energy surface of the EPI-002 and EPI-7170 bound ensembles as a function of the number of intramolecular contacts between aromatic residues and the α-helical order parameter Sα and provide illustrative examples of the conformational heterogeneity of bound conformations in Figs. S20, S21.

The EPI-002 and EPI-7170 bound states of Tau-5$_{R2\_R3}$ have relatively large hydrophobic cores, defined by a large number of contacts between aromatic residues. These hydrophobic cores consist of many different combinations of residues and vary substantially in size among bound conformations (Figs. S20, S21). The packing of these

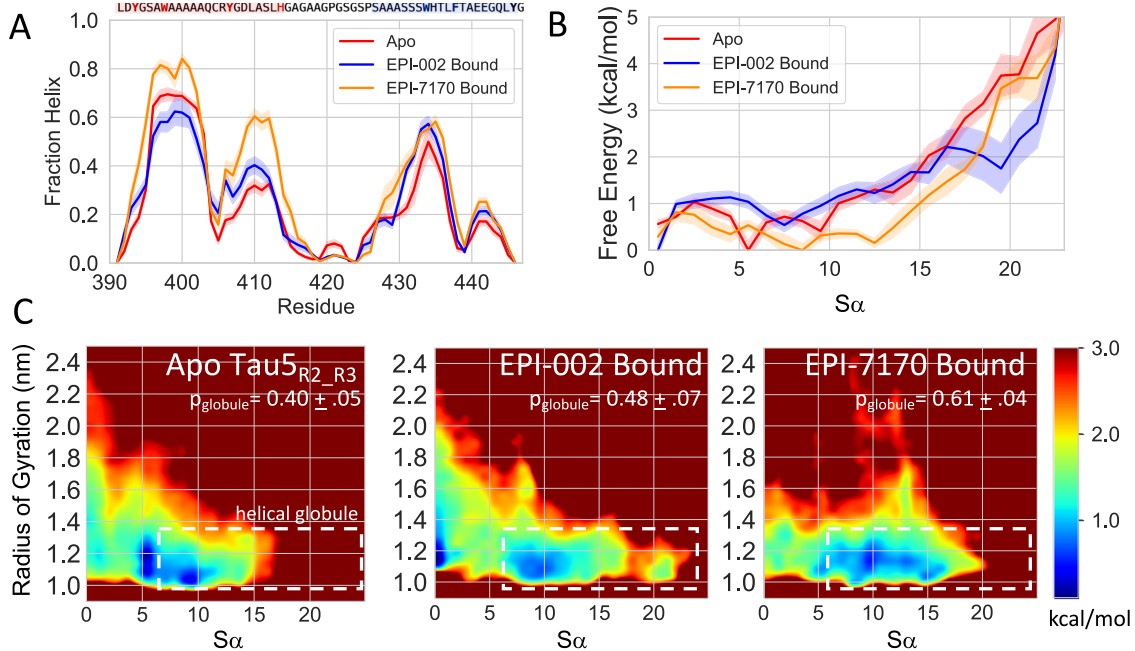

**Fig. 3 | EPI-002 and EPI-7170 binding induces the formation of collapse helical states of the Tau-5$_{R2\_R3}$ region of the androgen receptor N-terminal transactivation domain. A** Helical propensities were observed in the 300 K replica of explicit solvent REST2 MD simulations of Tau-5$_{R2\_R3}$ in its apo form (red) and in bound conformations obtained from simulations run in the presence of EPI-002 (blue) and EPI-7170 (orange). Simulated helical propensities are presented as mean values ± statistical error estimates from blocking. **B** Free energy surface of Tau-5$_{R2\_R3}$ conformations at 300 K as a function of the helical collective variable Sα for each ensemble. Sα describes the similarity of all consecutive six-residue fragments to ideal helical geometries. Six-residue fragments with an RMSD <0.5 Å from a canonical helix contribute a value of 1 to Sα, while six-residue fragments with an RMSD >3.0 Å contribute a value of 0 to Sα. Free energies were calculated using all frames of MD trajectories, and shaded regions indicate the standard error of the calculated free energy when the trajectory is split into five equally sized blocks. **C** Free energy surfaces as a function of the radius of gyration (R$_g$; reported in nm) and Sα. The dotted white lines indicate the defined boundary of the "helical globule" state (Sα > 6.0, R$_g$ < 1.3 nm). The population of the helical globule state is reported as p$_{globule}$. Source data are provided as a Source Data file.

hydrophobic cores is not well defined, and the relative orientations of aromatic sidechains vary substantially between bound conformations. These compact states also have a substantial population of helical conformations, and the locations of these helical elements and their relative orientations differ substantially among bound conformations. The structural properties of the ligand-bound conformations of Tau-5$_{R2\_R3}$ are consistent with the properties of partially folded molten-globule-like states observed protein folding[64–67] which have been observed to be more compact than a random coil and contain transiently populated helical conformations and dynamic clusters of aromatic and hydrophobic residues.

In simulations of Tau-5$_{R2\_R3}$ in the presence of ligands, we observe that bound conformations of EPI-002 and EPI-7170 have average R$_g$ values of 12.6 ± 0.3 Å and 12.4 ± 0.2 Å respectively, while the unbound conformations have average R$_g$ values of 13.9 ± 0.4 Å and 13.2 ± 0.2 Å respectively (Fig. 2B and Fig. S26). We note the unbound states of Tau-5$_{R2\_R3}$ from simulations in the presence of ligands have larger average R$_g$ values than states observed in the apo simulation of Tau-5$_{R2\_R3}$ (12.5 ± 0.1 Å). A direct comparison between these values is difficult, as EPI-002 and EPI-7170 have a higher affinity to compact conformations of Tau-5$_{R2\_R3}$ (Fig. S26) and the simulated timescales of association and dissociation events of EPI-002 and EPI-7170 to Tau-5$_{R2\_R3}$ will affect the populations of extended states observed in the lower solute temperature replicas of REST2 simulations. As a large fraction of frames in the simulations of Tau-5$_{R2\_R3}$ in the presence of ligands are bound, the apo simulation of Tau-5$_{R2\_R3}$ more extensively samples unbound states that are unperturbed by transient interactions with ligands. We, therefore, expect that the conformational properties of Tau-5$_{R2\_R3}$ observed in the apo REST2 MD simulation provide a more relevant comparison to the conformational properties of the ligand-bound ensembles.

To better understand the effects of ligand binding on the conformations of Tau-5$_{R2\_R3}$ we compare the helical propensity, R$_g$, and the Sα helical order parameters of the EPI-002 and EPI-7170 bound states to the apo Tau-5$_{R2\_R3}$ trajectory (Table 1 and Fig. 3). We observe that differences in the per-residue helical fractions of the apo Tau-5$_{R2\_R3}$ ensemble and the EPI-002:Tau-5$_{R2\_R3}$ bound ensemble are largely within statistical error estimates but the per-residue helical fractions of the EPI-7170:Tau-5$_{R2\_R3}$ bound ensemble are substantially higher than the apo and EPI-002 bound ensembles. We observe more pronounced differences between the Tau-5$_{R2\_R3}$ apo ensemble and the ligand-bound ensembles when considering differences in the Sα order parameter, which report on the cooperative formation of longer helical elements and the simultaneous formation of helical elements distant in sequence (See Methods). The free energy surface of the apo and ligand-bound ensembles of Tau-5$_{R2\_R3}$ are shown as a function of Sα (Fig. 3B) and as a function of R$_g$ and Sα (Fig. 3C). The apo ensemble of Tau-5$_{R2\_R3}$ has a pronounced free energy minimum centered at (Sα = 5, R$_g$ = 1.2 nm), a shallower minimum centered at (Sα = 9, R$_g$ = 1.1 nm) and no substantial minima with Sα > 10. The EPI-002 bound ensemble has a broad free energy minima centered at (Sα = 7.5, R$_g$ = 1.1 nm) and, in contrast to the apo ensemble, has an additional minimum ~1 kcal/mol higher in free energy centered at (Sα = 20.0, R$_g$ = 1.1 nm). The EPI-7170 ensemble is globally shifted to higher Sα regions relative to both ensembles, with a global minimum centered at (Sα = 9.0, R$_g$ = 1.1 nm) and a substantially elevated population of conformations with Sα > 10.0. This indicates that EPI-002 and EPI-7170 both stabilize the cooperative formation helical conformations in the R2 and R3 regions of Tau-5$_{R2\_R3}$, though EPI-7170 does so to a greater extent.

In order to quantify the relative populations of collapsed helical conformations in each ensemble, we define a "helical globule" state in each ensemble as all Tau-5$_{R2\_R3}$ conformation with Sα > 6.0 and

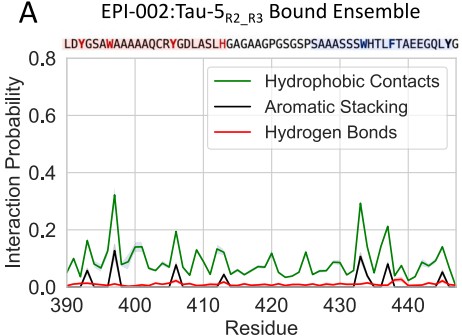

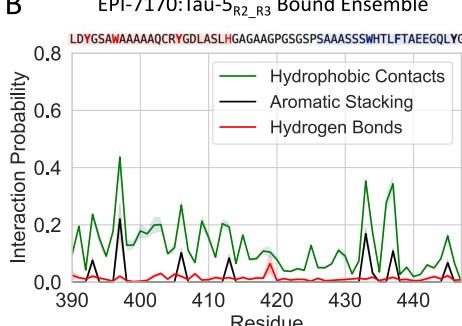

**Fig. 4 | Protein-ligand intermolecular interactions in the Tau-5$_{R2\_R3}$:EPI-002 and Tau-5$_{R2\_R3}$:EPI-7170 bound ensembles.** Populations of intermolecular interactions were observed in the Tau-5$_{R2\_R3}$:EPI-002 (**A**) and Tau-5$_{R2\_R3}$:EPI-7170 bound ensembles (**B**). Populations are calculated considering only the bound frames of R$_g$ <1.3 nm and report the populations, and relative free energies of this state ($\Delta G_{globule}$) at 300 K in each simulation (Table 2 and Fig. 3C).

MD simulations in the presence of ligands. Populations of intermolecular interactions are presented as mean values ± statistical error estimates from blocking. Source data are provided as a Source Data file.

R$_g$ <1.3 nm and report the populations, and relative free energies of this state ($\Delta G_{globule}$) at 300 K in each simulation (Table 2 and Fig. 3C). Conformations with S$\alpha$ > 6.0 correspond to Tau-5$_{R2\_R3}$ conformations where at least six 6-residue fragments have small RMSDs from an ideal helical conformation (See Methods). We observe that the helical globule state has a population of 40.4 ± 5.2% in the apo simulation, a modestly increased population in the EPI-002 bound ensemble (48.5 ± 6.5%) and a substantially larger population in the EPI-7170 bound ensemble (61.1 ± 4.5%). We note that relative populations of this state are sensitive to the selected threshold of S$\alpha$. We selected the threshold value of S$\alpha$ = 6.0 based on our desire to quantify the stability of ligand-bound conformations with more cooperative helical structure than the free energy minimum of the apo ensemble. We provide convergence plots of the simulated values of S$\alpha$ and the helical globule populations and compare the calculated populations of the helical globule state as a function of the selected cutoff value of S$\alpha$ in Fig. S27. We observe that the helical globule populations of ligand-bound states increase relative to the helical globule populations of the Tau-5$_{R2\_R3}$ apo ensemble as larger values S$\alpha$ are used as a cutoff.

**Aromatic stacking interactions of the dichlorinated phenyl ring of EPI-7170 stabilize compact helical conformations of Tau-5$_{R2\_R3}$**

We conducted a detailed dissection of the intermolecular interactions between Tau-5$_{R2\_R3}$ and the EPI compounds to better understand the molecular features of EPI-7170 that confer tighter binding and stabilize collapsed helical conformations of Tau-5$_{R2\_R3}$. The populations of intermolecular hydrophobic contacts, hydrogen bonds, and aromatic stacking interactions with each residue of Tau-5$_{R2\_R3}$ are shown for both ligands in Fig. 4 (See Methods). We observe relatively similar hydrophobic contact profiles between the two ligands, with the exception of elevated hydrophobic contact probabilities between EPI-7170 and $^{436}$LF$^{437}$.

The most significant differences in the binding modes of EPI-002 and EPI-7170 are the elevated populations of aromatic stacking interactions in the EPI-7170 bound ensemble (See Methods). The stacking populations of each aromatic residue in the EPI-002 and EPI-7170 bound ensembles are shown in Fig. 5A. The largest increases in stacking propensity in the EPI-7170 bound ensemble relative to the EPI-002 bound ensemble to occur in residues W397 (14.81 ± 0.04% vs 5.46 ± 0.02% population) and W433 (11.34 ± 0.01% vs 4.60 ± 0.0 1%). All aromatic residues experience a 2–3-fold increase in stacking propensity in the EPI-7170 bound ensemble relative to the EPI-002 bound ensemble. We examined the orientations of the stacking interactions occurring in both ensembles, calculating the propensity of each ligand ring to form face-to-face parallel stacking interactions and T-shaped stacking interactions

with each aromatic residue (Fig. 5 and Figs. S28,S29). We find that the dichlorinated phenyl ring of EPI-7170 forms substantially more parallel stacking interactions than the unchlorinated EPI-7170 phenyl ring and both phenyl rings of EPI-002 (Figs. S25, 26). We also observe that the unchlorinated phenyl rings of EPI-002 and EPI-7170 sample a broader distribution of ring orientations than the chlorinated ring of EPI-7170, with fewer well-defined free energy minima.

We observe an ~2-fold increase in hydrogen bond population between EPI-7170 and the R2 residues $^{403}$QCRYGD$^{408}$ relative to the hydrogen bond populations observed in the EPI-002 bound ensemble. The average hydrogen bond propensity for these residues is 2.1 ± 0.4% in the EPI-7170 bound ensemble compared to 1.3 ± 0.2% in the EPI-002 bound ensemble (Fig. 4 and Fig. S27). The most populated hydrogen bond interactions occur with Q403, R405, and D408. The populations of Q403, R405, and D408 hydrogen bonds are 3.0 ± 0.3%, 2.7 ± 0.6%, and 2.9 ± 0.5% in the EPI-7170 bound ensemble and 1.4 ± 0.2%, 1.4 ± 0.2%, and 1.2 ± 0.2% in the EPI-002 bound ensemble, respectively. For each of these residues, there are many distinct hydrogen bond pairs with similar populations. These hydrogen bonds contain both sidechain and backbone atoms functioning as donors and acceptors with different ligand atoms in the alkyl chains of EPI-002 and EPI-7170. In general, we cannot identify small subsets of dominant hydrogen bonding pairs in any region of Tau-5$_{R2\_R3}$ with the exception of a relatively highly populated hydrogen bond between the backbone amide of G419 and the oxygen atom in the chlorohydrin group of EPI-7170 (6.4 ± 3.4%). This hydrogen bond is predominantly populated in a metastable set of bound conformations sampled in a contiguous 900 ns portion of the trajectory, and correspondingly its population has a larger statistical error estimate than all other hydrogen bond populations. W397 hydrophobic contacts and aromatic stacking interactions are formed in the majority of bound frames containing this hydrogen bond (95 and 75%, respectively), illustrating that this interaction is predominantly present in a relatively narrow subset of bound conformations compared to more dynamic intermolecular interactions that are observed in more conformationally diverse sub-ensembles of bound conformations.

We observe that residues with the largest hydrogen bond propensities (Q403, R405, and D408) all have relatively high conditional interaction probabilities with hydrophobic and aromatic stacking interactions in other regions of Tau-5. W397 hydrophobic interactions occur in ~50% of frames containing hydrogen bonds between EPI-7170 and Q403, and W397 aromatic stacking interactions occur in ~25% of frames containing hydrogens bonds with Q403. F437 hydrophobic and aromatic stacking interactions occur in ~50 and ~25% of frames containing hydrogen bonds between EPI-7170 and R405, respectively, and Y445 hydrophobic and aromatic interactions occur in ~70% and ~50%

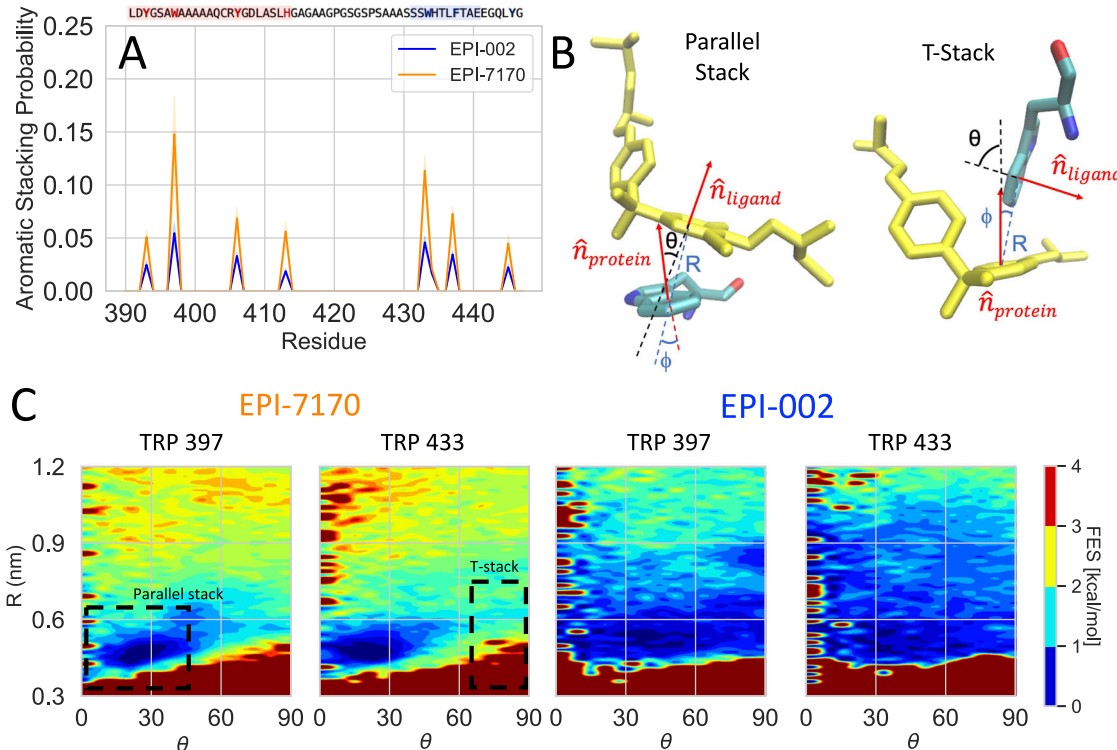

**Fig. 5 | Aromatic stacking interactions in the Tau-5$_{R2\_R3}$:EPI-002 and Tau-5$_{R2\_R3}$:EPI-7170 bound ensembles. A** Populations of aromatic stacking contacts between aromatic sidechains of Tau-5$_{R2\_R3}$ and aromatic rings of EPI-002 and EPI-7170 were observed in the 300 K replica of explicit solvent REST2 MD simulations. Populations are presented as mean values ± statistical error estimates from blocking. **B** Definition of angles and distances used to define stacking orientations and examples of parallel stacked and T-stacked conformations between EPI compounds and a tyrosine sidechain. For a protein aromatic ring and a ligand aromatic ring, R is defined as the distance between ring centers, $\hat{R}$ is defined as the unit vector connecting the ring centers, $\hat{n}_{protein}$ and $\hat{n}_{ligand}$ are normal vectors to the sidechain and ligand ring planes originating from the ring centers, θ is the angle between $\hat{n}_{protein}$ and $\hat{n}_{ligand}$, and φ is the angle between $\hat{n}_{protein}$ and $\hat{R}$. Parallel stacked conformations are defined as occurring when R < 6.5 Å, θ < 45°, & φ <60° and T-stacked conformations are defined as occurring when R < 7.5 Å, θ > 75°, & φ <60°. **C** Free energy surfaces as a function of R and θ between the chlorinated phenyl ring of EPI-7170 and the equivalent non-chlorinated phenyl ring in EPI-002 and the indole rings of TYR 397 and TYR 433. Source data are provided as a Source Data file.

of frames containing hydrogen bonds between EPI-7170 and D408, respectively. These high conditional interaction probabilities suggest the increased hydrogen bond populations observed in the EPI-7170 bound ensemble relative to the EPI-002 bound ensemble result from the dichlorinated phenyl ring of EPI-7170 more effectively localizing the ligand in the aromatic core of Tau-5$_{R2\_R3}$ through its increased hydrophobicity and planar aromatic stacking propensity. Once EPI-7170 is buried in helical globule states of Tau-5$_{R2\_R3}$ more hydrogen bonds likely become accessible through small displacements in ligand positions via a dynamic shuttling mechanism[32]. We note that the conditional interaction probabilities between hydrogen bonds and hydrophobic and aromatic interactions with non-neighboring residues are substantially higher than those observed in simulations of lower affinity compounds binding to α-synuclein[32], suggesting that the more cooperative formation of intermolecular interactions observed in the EPI-002 and EPI-7170 bound ensembles to confer the higher affinity binding observed in this investigation.

**Non-covalent binding of EPI-7170 and EPI-002 increases the proximity of their weakly reactive chlorohydrin groups to the nucleophilic thiol of CYS404 in collapsed helical states of Tau-5$_{R2\_R3}$**

EPI-002 and EPI-7170 both possess a chlorohydrin group (Fig. 2). The chlorohydrin group of EPI-002 has been shown to be weakly reactive with cysteine residues in the AR-NTD, and it is hypothesized that covalent attachment of EPI compounds to Tau-5 may be required for its biological activity[14,37,38]. It has previously been proposed that fast reversible non-covalent interactions of EPI compounds to different

regions of the AR-NTD may drive covalent attachment to specific cysteines in AR[38]. Based on previously reported NMR experiments on the AR-NTD that showed the largest backbone amide NMR CSPs in the R2 and R3 regions[24] in EPI-002 titrations, we hypothesize that non-covalent binding of EPI ligands may direct covalent attachment of these ligands to CYS404. CYS404 is found in the middle of the transiently helical R2 region ($^{397}$WAAAAAQCRYG$^{407}$) before the helix-breaking GLY407 residue. If covalent reactivity at CYS404 is important for the biological activity of EPI ligands, we suspect that EPI ligands with non-covalent binding modes that more effectively localize their weakly reactive chlorohydrin groups to the nucleophilic thiol of CYS404 will be more potent AR inhibitors in vivo. We compare the distance distributions between the chlorohydrin chlorine atoms of EPI-7170 and EPI-002 to the sulfur atom of CYS404 of Tau-5$_{R2\_R3}$ in our ligand binding simulations in Fig. S31. We find that the higher non-covalent affinity of EPI-7170 relative to EPI-002 dramatically increases the proximity of its chlorohydrin group to CYS404 relative to the distance distribution observed between the chlorohydrin group of EPI-002 and CYS404. We find that the chlorohydrin chlorine atom of EPI-7170 is within 10.0 Å of the CYS404 thiol sulfur atom in 23.0% of simulation frames, while the chlorine atom of EPI-002 is within 10.0 Å of the CYS404 thiol sulfur atom in 9.6% of simulation frames. We note that in addition to increasing the proximity of the EPI ligand chlorohydrin groups to CYS404, the compact nature of the ligand-bound ensembles may further facilitate covalent attachment by sequestering the reaction from solvent water molecules. The conformational properties of covalent adducts of EPI compounds bound to Tau-5 will be the subject of future investigations.

## Binding mode comparisons of EPI-002 and EPI-7170 with Iodo-EPI-002 and BADGE:2H₂O

To obtain additional insight into the inhibition mechanisms of EPI-002 and EPI-7170, we also conducted REST2 MD simulations of Tau-5$_{R2\_R3}$ in the presence of two additional ligands: Bisphenol-A Diglycidic Ether:2H$_2$O (BADGE:2H$_2$O), and Iodo-EPI-002 (Fig. S32). BADGE:2H$_2$O has an identical structure to EPI-002 aside from the replacement of the EPI-002 chlorohydrin group with a diol. This substitution eliminates the covalent reactivity of the compound, and cellular assays have shown that BADGE:2H$_2$O does not inhibit AR transcriptional activity[37,38]. Iodo-EPI-002 differs from EPI-002 only be the addition of a single iodine atom on the bisphenol-A phenyl ring closest to the chlorohydrin group has and has been shown to be an ~10x more potent AR inhibitor in cellular AR transcriptional activity inhibition assays[39]. A REST2 simulation of BADGE:2H$_2$O and Tau-5$_{R2\_R3}$ was run for 4.2 μs/replica, for an aggregate simulation time of 67 μs, and a REST2 simulation of Iodo-EPI-002 and Tau-5$_{R2\_R3}$ was run for 5.0 μs/replica, for an aggregate simulation time of 80 μs. The simulated properties of Tau-5$_{R2\_R3}$ observed in these simulations are compared to the apo Tau-5$_{R2\_R3}$ simulations in Table S1 and Figs. S32, S33. Simulation and convergence analyses for these simulations are reported in Figs. S32, S33.

We observe that BADGE:2H$_2$O has a simulated $K_D$ of 4.94 ± 0.32 mM, which is within statistical uncertainty estimates of the simulated K$_D$ values of EPI-002 (5.24 ± 0.43 mM). We observe that most simulated properties are within statistical uncertainty estimates of the simulated properties of EPI-002 (Table S1 and Figs. S32, S33) and that the intermolecular interactions between Tau-5$_{R2\_R3}$ and each ligand observed in the bound ensembles are very similar (Fig. S34). The similar simulated binding affinities and properties of the bound ensembles of EPI-002 and BADGE:2H$_2$O are consistent with the hypothesis that differences in their biological potency may strictly result from a lack of covalent reactivity, rather than from differences in their non-covalent binding affinity or non-covalent binding mechanisms. These results are consistent with the hypothesis that covalent attachment is essential for the biological activity of EPI AR-NTD inhibitors.

Iodo-EPI-002 has been shown to be substantially more potent than EPI-002 in cellular AR inhibition assays[39]. Iodo-EPI-002 was shown to have IC$_{50}$ values of ~1 μM in prostate-specific antigen luciferase reporter assays for inhibition of endogenous AR transcriptional activity[39] compared to IC$_{50}$ values of ~10 μM for EPI-002 in similar assays[37,38]. EPI-7170 was also found to have an IC$_{50}$ value of ~1 μM in cellular prostate-specific antigen luciferase reporter assays[44]. We found that the simulated $K_D$ of Iodo-EPI-002 (1.89 ± 0.13 mM) is very similar to the simulated $K_D$ of EPI-7170 (1.92 ± 0.15 mM). Interestingly, we observe that the Iodo-EPI-002:Tau-5$_{R2\_R3}$ bound ensemble has a larger fraction helix (38.4 ± 1.7%) and helical globule population (79.2 ± 3.3%) compared to the EPI-7170:Tau-5$_{R2\_R3}$ bound ensemble (fraction helix of 32.8 ± 0.5% and helical globule population of 61.1 ± 4.5%). We observe that the populations of intermolecular interactions in the ligand Iodo-EPI-002:Tau-5$_{R2\_R3}$ bound ensembles are very similar to those observed in EPI-7170:Tau-5$_{R2\_R3}$ bound ensemble (Fig. S34). These simulation results suggest that Iodo-EPI-002 and EPI-7170 bind Tau-5 with similar binding mechanisms and similar affinities and are consistent with the similar potencies observed in cellular AR transcriptional inhibition assays.

## Discussion

The design of small molecule inhibitors targeting the disordered N-terminal transactivation domain (NTD) of the androgen receptor is a promising avenue for the development of drugs to treat castration-resistant prostate cancer (CPRC) and an important proof-of-principle of the feasibility of IDP drug-design. Recent preclinical results for EPI-7386[44], a second-generation AR-NTD inhibitor from the EPI-7170 compound family which entered human trials in 2020 (https://clinicaltrials.gov/ct2/show/NCT04421222), are encouraging and suggest that EPI family compounds are viable CRPC drug candidates. A large amount of biophysical and biological research has been carried out to characterize the interactions of EPI compounds with the AR-NTD and their therapeutic potential, but until now, no atomic resolution structural models have existed to explain the molecular mechanism of AR-NTD inhibition or rationalize differences in the activity of compounds in the EPI family.

In this investigation, we have leveraged recent advances in molecular simulation force fields and enhanced sampling methods to provide atomic resolution binding mechanism models of EPI-002 and EPI-7170 that are highly consistent with previously reported NMR measurements[24]. We utilized long-timescale enhanced sampling MD simulations with the state-of-the-art a99SB-disp force field[47] to study the conformational properties of Tau-5$_{R2\_R3}$, a disordered fragment of the AR-NTD that contains the Tau-5 residues that showed the most pronounced NMR chemical shift perturbations in EPI-002 titrations[24]. We found that the apo ensemble of Tau-5$_{R2\_R3}$ obtained from MD simulations is in excellent agreement with previously measured NMR chemical shift data[24]. Quantitative comparisons with backbone chemical shifts reveal relatively minor discrepancies in the simulated helical propensity of the R3 region, which can be corrected using a maximum-entropy ensemble reweighting algorithm[59,60]. The apo ensemble of Tau-5$_{R2\_R3}$ reveals substantially populated intramolecular contacts between the R2 and R3 regions, which were previously hypothesized to be important for EPI-002 binding[24]. Simulations of Tau-5$_{R2\_R3}$ in the presence of EPI-002 revealed that EPI-002 binds via a dynamic and heterogenous ensemble of interconverting binding modes and does not induce the formation of a stable folded conformation of Tau-5$_{R2\_R3}$, an observation that is consistent with relatively small NMR CSPs observed in ligand titrations[24]. We find that EPI-002 binding is predominantly driven by interactions with aromatic residues in the transiently formed R2 and R3 helices, and that these helical regions can "wrap around" EPI-002 to form compact helical conformations that resemble dynamic molten-globule states from protein folding, which we define as a helical globule state.

The compound EPI-7170 has a 4-6-dichloro substituted phenyl ring in its bisphenol-A scaffold and a methylsulfonamide group in place of a hydroxyl group in an alkyl chain relative to EPI-002. MD simulations of Tau-5$_{R2\_R3}$ in the presence of EPI-7170 reveal that EPI-7170 has an ~2.5-fold higher affinity to Tau-5$_{R2\_R3}$ than EPI-002. We observe that EPI-7170 has a substantially higher probability of simultaneously interacting with residues in both the R2 and R3 helices in its bound ensemble. EPI-7170 binding substantially increases the helical populations of Tau-5$_{R2\_R3}$ relative to the apo Tau-5$_{R2\_R3}$ ensemble and the EPI-002:Tau-5$_{R2\_R3}$ bound ensemble. Analysis of the α-helical order parameter (Sα) reveals that EPI-002 and EPI-7170 both induce the cooperative formation of helical elements in the R2 and R3 regions and that the EPI-7170:Tau-5$_{R2\_R3}$ bound ensemble has a substantially elevated population of helical globule conformations relative to the apo Tau-5$_{R2\_R3}$ ensemble and EPI-002:Tau-5$_{R2\_R3}$ bound ensemble. Our simulations identify that the increased affinity of EPI-7170 results from an increased propensity of the dechlorinated phenyl ring to form parallel face-to-face stacking interactions with aromatic sidechains relative to the phenyl rings EPI-002. These stacking interactions localize EPI-7170 to a dynamic hydrophobic/aromatic core of Tau-5$_{R2\_R3}$, where it adopts an increased propensity to form an array of interconverting hydrogen bonding interactions with residues in the R2 region. We observe a very similar binding affinity and binding mechanism for Iodo-EPI-002, a ligand that only differs from EPI-002 by the addition of a single iodine atom on a bisphenol-A phenyl ring that has been shown to have similar potency to EPI-7170 in cellular AR transcriptional inhibition assays[37–39,44]. These results underscore the importance of increased aromatic

stacking propensities in bisphenol-A phenyl rings for increasing the affinity of EPI compounds to Tau-5.

NMR measurements of the interaction between EPI-7170 and Tau-5 have yet to be reported, which may be the result of the relative insolubility of EPI-7170 relative to EPI-002[42]. A number of small molecule ligands that bind IDPs have been found to be relatively insoluble at concentrations required for biophysical experiments, in particular NMR spectroscopy[32]. Obtaining sufficiently soluble small molecules of drugs is often a challenge in drug discovery campaigns targeting folded proteins, where active ligands can have very tight picomolar-nanomolar affinities. As many small molecule disordered proteins ligands with biological activity have been found to have relatively low μM-mM affinities in biophysical assays, obtaining sufficient solubility for detailed biophysical characterization of small molecule ligands may be particularly challenging in drug discovery campaigns for disordered targets. This suggests that solubility may be a ubiquitous problem when studying small molecules that bind IDPs and underscores the value of combining insights from experimental and computational studies when studying compounds near the detectable solubility limits of biophysical experiments. In this investigation, we have accessed the accuracy of our simulations of apo Tau-5$_{R2\_R3}$ and EPI-002 Tau-5$_{R2\_R3}$ binding by comparisons with previously measured NMR data. The consistency between our simulations and NMR data suggests that the simulation methods and force fields utilized are well suited to describe the conformational properties of Tau-5$_{R2\_R3}$ and the interactions between Tau-5$_{R2\_R3}$ and EPI compounds with a common bisphenol-A scaffold. Given the relatively small differences in the chemical structures of EPI-002 and EPI-7170 and the quality of ligand molecular mechanics force fields[32,61], we expect that our simulation model will be capable of discerning differences in their binding modes as observed in a previous study[32].

While NMR data has not yet been reported for interactions between EPI-7170 and Tau-5$_{R2\_R3}$, protein and ligand-detected NMR data have demonstrated direct interactions between Tau-5 and EPI-002 and Tau-5 and the related compound EPI-7386[44]. NMR studies of EPI-7386 have identified NMR CSPs of the sidechain resonances of W397 and W433 of Tau-5 in EPI-7386 titrations, which is consistent with the importance of these residues for EPI compound binding in our simulations. Saturation transfer difference (STD) experiments also confirm direct interactions between EPI-7386 and Tau-5, demonstrating that these CSPs do not only result from allosteric changes in the conformational ensemble of Tau-5 upon EPI-7386 binding[44]. We hypothesize that as EPI-7170 shares a common Bisphenol-A scaffold with a large number of EPI derivatives that have been shown to have in vitro and in vivo interactions with AR and EPI-7170 has been shown to have a similar biological activity to these ligands[37–39,44] that EPI-7170 also directly binds the Tau-5 region of AR. The relative insolubility of EPI-7170 may make biophysical measurements to confirm direct AR binding difficult to perform and interpret, but the further biophysical characterization of the binding modes of more soluble ligands that bind the Tau-5 region of AR are of great interest and will be the subject of future investigations.

The chlorohydrin group of EPI-002 has been found to be weakly covalently reactive with Tau-5, and it is hypothesized that covalent attachment of EPI compounds to Tau-5 is required for their biological activity[14,37–39]. We have simulated Tau-5$_{R2\_R3}$ binding the compound BADGE:2H$_2$O, which replaces the chlorohydrin group EPI-002 with an unreactive diol. We observe that EPI-002 and BADGE:2H$_2$O have very similar simulated binding affinities, and that the EPI-002:Tau-5$_{R2\_R3}$ and BADGE:2H$_2$O:Tau-5$_{R2\_R3}$ bound ensembles are very similar, in agreement with the hypothesis that covalent reactivity is essential for AR-NTD inhibition in cellular contexts. In our simulations, we observe that reversible non-covalent binding of EPI-002 and EPI-7170 at the dynamic interface between the R2 and R3 regions of Tau-5 increases the proximity of their weakly reactive chlorohydrin groups to the thiol

group of cysteine 404, and that the higher affinity binding of EPI-7170 more effectively localizes its chlorohydrin group to the cysteine 404 thiol. We hypothesize that the non-covalent affinity of EPI compounds to this region may drive preferential covalent attachment to this cysteine residue relative to other cysteine residues in the AR-NTD. In this scenario, we expect that compounds with higher non-covalent binding affinity to Tau-5$_{R2\_R3}$, such as EPI-7170 and Iodo-EPI-002, will be more covalently reactive in cellular contexts. The conformational properties of covalent adducts of EPI compounds bound to Tau-5 will be the subject of future investigations.

Molecular simulation studies of the interactions between small molecules and IDPs are becoming more common[30,32–36], but simulation studies that provide detailed comparisons of the binding modes of multiple small molecules and identify subsets of intermolecular interactions that confer differences in their specificity and affinity have only recently begun to emerge[32]. Understanding the molecular features of dynamic ligand binding modes that confer affinity and specificity among known IDP ligands is an essential step in developing rational design strategies to improve the affinity of small molecules to IDPs or to design inhibitors de novo for new IDP sequences. Identifying the key intermolecular interactions that underpin IDP ligand binding modes enables medicinal chemists to identify promising regions of chemical space to explore for further ligand derivatization and provides experimentally testable hypotheses to validate and refine binding mechanism models in drug discovery campaigns. In this investigation, we have identified the importance of aromatic stacking orientations of the phenyl groups of the Bisphenol-A scaffold and hydrogen bonding interactions in the alkyl chains of EPI compounds for stabilizing compact helical globule states of Tau-5$_{R2\_R3}$. These mechanistic insights suggest important small molecule features to explore in the development of more potent androgen receptor inhibitors for the treatment of CRPC.

The simulations conducted in this study were carried out on a fragment of the Tau-5 region of AR containing the residues previously observed to have the highest propensity to interact with EPI-002 to obtain a computationally tractable system and enable statistically meaningful comparisons of the binding modes of EPI-002 and EPI-7170. We note that the R1 helix of Tau-5 was also found to interact with EPI-002 by NMR in the context of the full Tau-5 region but that a peptide containing only the R1 domain was not found to interact with EPI-002 at similar concentrations. Based on the sequence composition of R1, in particular the presence of 6 aromatic residues in a 31-residue domain and the helical propensity observed by NMR, we speculate that EPI-002 and EPI-7170 likely interact with the R1 region with similar bindings mechanisms to those observed in this investigation, and that EPI-7170 transiently binds to R1 with a higher affinity than EPI-002. We hypothesize that EPI compounds stabilize transient long-range intramolecular between the R1 and R2 regions, R1 and R3 regions, and simultaneous interactions between the R1, R2 and R3 regions with similar binding modes to those observed between the R2 and R3 regions in this investigation. Due to the presence of the relatively rigid polyproline linker between R1 and R2, we suspect that interactions involving the R1 helix are less important for conferring the affinity of EPI compounds to Tau-5 than the interactions between R2 and R3 described here. Studying the interactions of EPI compounds with larger constructs of Tau-5 that contain the R1, R2, and R3 regions will require substantially longer simulations than those reported here, and in practice, will likely require the optimization of additionally enhanced sampling algorithms for this system.

Finally, we note that the three cysteine residues in the Tau-5 region of AR and four additional cysteine residues in the AR-NTD may also form disulfide bonds in certain cellular conditions. Additionally, several post-translation modifications which substantially alter the cellular activity AR, such as phosphorylation at Serine 424 in the R3 helix, have been discovered[50,68–70]. The conformational ensembles

presented here, along with future computational and experimental studies of the conformational ensembles of post-translationally modified Tau-5 constructs and Tau-5 covalent adducts, may be helpful for understanding the impact of these modifications on AR function in vivo.

## Methods

### Simulation methods

Simulations of the androgen receptor Tau-5$_{R2\_R3}$ region (residues L391-G446, capped with ACE and NH2 groups) in the presence and absence of ligands were performed using GROMACS 2019.2[71,72] patched with PLUMED v2.6.0[73]. The AR Tau-5$_{R2–R3}$ protein parameterized using the a99SB-*disp* force field and water molecules were parameterized with the a99SB-disp water model[47]. Simulations with ligands were run using GAFF1[61] for ligand forcefield parameters, obtained from ACPYPE[74]. Each system was solvated with 13200 water molecules in a cubic box with a length of 7.5 nm and neutralized with a salt concentration of 20 mM NaCl by 8 Na$^+$ ions and 5 Cl$^−$ ions. Energy minimization of the system is performed with the steepest descent minimization algorithm to the maximum force smaller than 1000.0 kJ/(mol/nm). Equilibration was first performed in the NVT ensemble for 2000 ps at the temperature of 300 K using the Berendsen thermostat[75] then the systems were further equilibrated in the NPT ensemble for 200 ps at a target pressure of 1 bar with the temperature at 300 K maintained by Berendsen thermostat, with position restraints added to all heavy atoms. Bond lengths and angles of protein and ligand atoms were constrained with the LINCS[76] algorithm and water constraints were applied using the SETTLE algorithm[77]. Canonical sampling in the NVT ensemble algorithms was obtained using the Bussi et. al. velocity rescaling thermostat[78] with a 2 fs timestep. The PME algorithm[79] was utilized for electrostatics with a grid spacing of 1.6 nm. Van der Waals forces were calculated using a 0.9 nm cut-off length. The REST2 algorithm[51,80] was utilized with an exchange attempted every 80 ps, selecting all protein atoms as the solute region. A 16-replica temperature ladder ranging from 300–500 K was utilized to scale the solute temperature. Apo Tau-5$_{R2–R3}$, Tau-5$_{R2–R3}$ + EPI-002, Tau-5$_{R2–R3}$ + EPI-7170, Tau-5$_{R2–R3}$ + BADGE:2H$_2$O, and Tau-5$_{R2–R3}$ + Iodo-EPI-002 were simulated for 4.6, 4.0, 4.5, 4.2, and 5.0 μs per replica respectively, for total simulation times of 74, 64, 72, 67, and 80 μs respectively. Frames were saved every 80 ps for analysis.

Initial structures of Tau-5$_{R2\_R3}$ were generated with the pmx software[81]. The builder module of the pmx software package was used to generate a fully helical Tau-5$_{R2\_R3}$ conformation with all residues in an ideal helical conformation ($\phi = −57, \psi = −47$) and a partially helical Tau-5$_{R2\_R3}$ conformation where only residues in the R2 and R3 regions were built in helical conformations and the remaining residues were left in an extended conformation. These starting structures were subject to an energy minimization, and then each structure was used to perform a short 100 ps 600 K high-temperature unfolding simulation in vacuo in the NVT ensemble. Eight structures were selected from each of the unfolding trajectories to span a range of helicity in starting structures. Initial conformations of EPI-002, EPI-7170, Iodo-EPI-002, and BADGE:2H$_2$O were generated using the Open Babel online toolbox[82]. Starting structures for Tau-5$_{R2\_R3}$ simulations in the presence of ligands were prepared by inserting the ligands into simulation boxes with the same 16 Tau-5$_{R2\_R3}$ starting structures and repeating the process of solvation, neutralization, energy minimization, and NVT and NPT equilibration as described above. Secondary structure populations were calculated from MD trajectories using the DSSP algorithm[83]. Analyses were run utilizing MDtraj[84] and the numpy[85] python package.

### Statistical error estimates

Statistical error estimates of the simulated properties from MD simulations were calculated using a blocking analysis following ref. 52 with an optimal block size selection determined as described by ref. 53, using the *pyblock* python package. In this procedure, the trajectory is divided into a given number of equally sized "blocks", average values of simulated quantities are computed for each block, and the standard error of the average values calculated across all blocks is used as an error estimate. Optimal block size is selected to minimize the estimated error of the standard error across blocks, according to ref. 53.

### Maximum-entropy ensemble reweighting

NMR chemical shifts were calculated for the 300 K base replica of the REST2 MD simulation of apo Tau-5$_{R2\_R3}$ using SPARTA+[55]. We utilized the maximum-entropy reweighting algorithm of ref. 59,60 with Cα NMR chemical shifts as restraints. We utilized a gaussian error model for Cα chemical shift predictions with a standard deviation $\sigma = 1.73$ ppm. The Kish ratio[86,87] of the resulting ensemble was 43.9, which means that the algorithm effectively retained 43.9% of the frames from the unbiased simulation in the reweighted ensemble.

### Sα α-helical order parameter

The α-helical order parameter $S\alpha$, measures the similarity of all six-residue segments to an ideal helical structure ($\phi = −57, \psi = −47$)[54]. $S\alpha$ is calculated according

$$S_\alpha = \sum_i^N \frac{1 - \left(\frac{\text{RMSD}\alpha_i}{r_0}\right)^8}{1 - \left(\frac{\text{RMSD}\alpha_i}{r_0}\right)^{12}} 1$$

where the sum is over $N$ consecutive six-residue segments, RMSDα$_i$ is the Cα-RMSD between an ideal α-helical geometry a six-residue fragment (spanning from residue $i$ to residue $i+5$), and $r_0 = 1.0$ Å. When $r_0 = 1.0$ Å, a six-residue fragment with a value of RMSD$_\alpha$ <0.5 Å contributes a value of ~1 to the $S_\alpha$ sum, a six-residue fragment with a value of RMSD$_\alpha$ = 1.1 Å contributes a value of ~0.5 to the $S_\alpha$ sum, and a six-residue fragment with a value of RMSD$_\alpha$ >3.0 Å contributes a value of ~0 to the $S_\alpha$ sum. The value of $S_\alpha$ for a protein conformation can therefore be interpreted as a proxy for the number of six-residue fragments closely resembling an ideal helical conformation. A completely helical conformation of the 56-residue Tau-5$_{R2\_R3}$ construct has an $S_\alpha$ value of 51, and a Tau-5$_{R2\_R3}$ with no helical content has an $S_\alpha$ value of 0.

### Ligand contacts and $K_D$ calculations

We define an intermolecular contact between a ligand and a protein residue as occurring in any frame where at least one heavy (non-hydrogen) atom of that residue is found within 6.0 Å of a ligand-heavy atom. To calculate a simulated $K_D$ value for each compound, we define the bound population (P$_b$) of each ligand as the fraction of frames with at least one intermolecular contact between a ligand and Tau-5$_{R2\_R3}$. We calculated the $K_D$ value according to $K_D = P_u/P_b(vc°N_A)^{−1}$ where (P$_u$) is the fraction of frames with no ligand contacts, $v$ is the volume of the simulation box, $c°$ is a standard state concentration (1 mol L$^{−1}$), and $N_A$ is Avogadro's number[63]. In the 7.5 nm simulation box used in this work containing one ligand and one protein molecule, $c° = 1$ mol L$^{−1}$, $v = 4.22 \times 10^{−25}$ L, and the concentration of the ligand and protein are each 3.93 mM.

### Ligand intermolecular interactions

Intermolecular hydrophobic contacts were defined as occurring when pairs of protein carbon and ligand carbon and protein carbon and ligand chlorine atoms were within 4 Å. Potential hydrogen bond donors were defined as all nitrogen, oxygen, or sulfur atoms with attached hydrogen, and potential hydrogen bond acceptors were defined as all nitrogen, oxygen, and sulfur atoms. Hydrogen bonds were identified with a distance cutoff of 3.5 Å between the donor-

hydrogen and heavy-atom acceptor, and a donor-hydrogen-acceptor angle >150°.

Aromatic stacking interactions were calculated following the conventions of Marsili et al.[82] (Shown in Fig. 5) with modified distance and angle cutoffs. The cutoffs were selected based on the observed locations of free energy minima of parallel (face-to-face) stacked and T-stacked conformations between the EPI ligands and protein aromatic sidechains. For a protein aromatic ring and a ligand aromatic ring: we define R as the distance between ring centers, $\hat{R}$ as the unit vector connecting the ring centers, $\hat{n}_{protein}$ and $\hat{n}_{ligand}$ are normal vectors to the sidechain and ligand ring planes originating from the ring centers, θ is the angle between $\hat{n}_{protein}$ and $\hat{n}_{ligand}$ (θ = arccos($|\hat{n}_{protein} \bullet \hat{n}_{ligand}|$)), and ϕ is the angle between $\hat{n}_{protein}$ and $\hat{R}$ (ϕ = arccos($|\hat{n}_{protein} \bullet \hat{R}|$)). Parallel stacked conformations were defined as occurring when R < 6.5 Å, θ < 60°, and ϕ <45° and T-stacked conformations were defined as occurring when R < 7.5 Å, θ > 75°, and ϕ <45°. In the evaluation of the free energy surfaces, we chose not to distinguish between angles of θ and π− θ and between angles of ϕ and (π−ϕ). Distance–angle probability distributions and corresponding free energy surfaces (Fig. 5) were normalized by a factor $R^2 \sin(θ)$ to obtain a flat free energy profile in the case of configurations with no angle or distance preference[88,89].

### Reporting summary
Further information on research design is available in the Nature Research Reporting Summary linked to this article.

## Data availability
The molecular dynamics trajectory data generated in this study has been deposited in Zenodo (https://doi.org/10.5281/zenodo.7120845) and is available on GitHub (https://github.com/paulrobustelli/AR_ligand_binding). The structural ensemble of apo Tau-5$_{R2\_R3}$ has been deposited in the Protein Ensemble Database[90] under deposition number PED00206. The androgen receptor Tau-5 NMR chemical shifts used for trajectory reweighting are deposited in Biological Magnetic Resonance Bank entry 51479. Source data are provided with this paper as Source Data File. Source data are provided with this paper.

## Code availability
All gromacs input files and code used for analyses are freely available from GitHub (https://github.com/paulrobustelli/AR_ligand_binding) and Zenodo (https://doi.org/10.5281/zenodo.7120845). Code for the pmx software[81] used to generate protein starting structures (https://github.com/deGrootLab/pmx) and the ACPYPE software[72] (https://github.com/alanwilter/acpype) used to generate ligand parameters is available on GitHub.

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

## Acknowledgements

This work was supported by the National Institutes of Health under award R35GM142750, the National Research Council supercomputing grant MCB200087P, AGAUR (2017 SGR 324), MINECO (PID2019-110198RB-I00), and the European Research Council (CONCERT, contract number 648201). J.Z. was supported by the China Scholarship Council (CSC ID: 201906320040) and R35GM142750. The authors thank Massimiliano Bonomi for assistance in implementing maximum-entropy reweighting methods, Matteo Paloni for providing scripts to calculate stacking interactions, and Stase Bielskute and Borja Mateos for stimulating discussions and critical reading of this manuscript. IRB Barcelona is the recipient of a Severo Ochoa Award of Excellence from MINECO.

## Author contributions

P.R and X.S. designed the research. J.Z. and P.R. performed MD simulations and data analysis. P.R. and J.Z. wrote the manuscript. X.S. participated in the discussion and revision of the manuscript. All authors reviewed and approved the final manuscript. P.R. supervised the project.

## Competing interests

Paul Robustelli is an Open Science Fellow and scientific consultant of Roivant Discovery and a scientific advisor and consultant of Dewpoint Therapeutics. Xavier Salvatella is a founder and board member of Nuage Therapeutics. These affiliations have not influenced this work. The remaining authors declare no competing interests.
