## [Peer Review File · Nature Communications]

REVIEWER COMMENTS

Reviewer #1 (Remarks to the Author):

EPI-002 and EPI-7170 are two compounds that were developed against the castration-resistant prostate cancer. These two compounds were shown to mainly bind to the R2 and R3 in the Tau-5 in the AR-NTD. Zhu et al. used molecular dynamics simulations to analyze the binding behavior of two compounds with a 56-residue segment containing R2 and R3. However, the simulated conformations contain obviously larger helical content than previous NMR results. Although the simulations showed that EPI-7170 might bind stronger than EPI-002, both of them are simulated to have K_d values in mM range, which are very weak. Compared to what is known before, this study does not offer much new insight, as it is already known that the compounds mostly bind to the R2 and R3 range and increase helical content. The other problem is that EPI-002 was known to form covalent adduct with the protein (ref. 24), which was not considered in the simulations.

Reviewer #2 (Remarks to the Author):

This paper describes an interesting study designed to elucidate how small compounds bind to an intrinsically disordered fragment of the androgen receptor (AR) using molecular dynamics (MD) simulations. This is an area of the utmost interest for research on prostate cancer because development of castration resistant prostate cancer is known to depend critically on the activity of the N-terminal transactivation domain of AR (AR NTD), which is intrinsically disordered. Small compounds that binding to this domain are currently in clinical trials to treat prostate cancer. Understanding how the compounds bind to the domain is therefore crucial to optimize the compounds and increase their potency, but defining the binding mode is strongly hindered by the weak affinity of the compounds and by the intrinsically disordered nature of the AR NTD. Although NMR spectroscopy provided some insights into the regions of the AR NTD that bind to some of the existing compounds, such as EPI-002, it is extremely difficult to determine the binding modes in atomic detail by any currently available experimental method.

Zhu et al. describe a commendable computational approach to shed light into how EPI-002 and a related more potent compound (EPI-7170) bind to the region of the AR NTD that was found to bind to EPI-002 by NMR spectroscopy. The MD simulations that they describe suggest that the compounds bind dynamically (i.e. in multiple modes) to the interface between two helices through networks of aromatic ring stacking interactions and hydrogen bonds. In all honesty, I am not sure to what extent the conformations and binding modes visited during the simulations represent the conformational ensembles and binding modes existing in solution. However, the observed binding modes make sense

from a chemical point of view and do involve the residues that exhibited large chemical shift changes upon addition of EPI-002 in previous NMR studies. Moreover, the authors make a strong effort to correlate the observed NMR chemical shifts with those predicted for the ensembles of conformations observed during the simulations. Hence, I believe that this is currently the best we can do to understand the binding modes. Overall, this study will be of strong interest to researchers on prostate cancer and also in general to those interested in binding of organic compounds to intrinsically disordered regions of proteins. I do have a few minor concerns that should be addressed before publication.

1. The authors do a reasonable job at not overselling their results and consider the limitations of their approach, but I would still encourage them to be careful with wording and tone down strong conclusions. As an example, I would prefer that the authors say that the NMR data 'are consistent with' or 'support the validity' of the MD simulation results rather than saying that the NMR data 'validate' the results.

2. Since one of the main interests of this study is the atomic detail information provided by the simulations on the interactions between the small compounds and the AR NTD, they should provide better visualization of these details. The structures shown in fig. 2D should be much larger (please protect people's eyes and do not assume that they can just expand the figure on a computer screen; figures should be readily evaluated on a printed copy). Moreover, they should show a representative gallery of the diverse binding modes observed in main figures (not in supplementary materials). The figures should allow the reader to observe how similar or different are the most populated binding modes observed for a given compound, and to observe the key interactions. Note that movies are useful, but cannot replace static pictures that can be studied calmly.

3. The criterion used to compute population of bound states, P_b , seems to be conceptually flawed, as binding is assigned simply from the proximity of two atoms from different molecules. If I understand correctly, this means that random encounters where two atoms come close in a transient manner, without staying close because of interactions between them, are considered as bound. Since the same criterion was used for the two compounds, the relative affinities measured for the two compounds may still be fine, but this limitation should be pointed out. Hopefully, the field will revise this methodology in the future.

4. In the discussion the authors state that solubility may be a ubiquitous problem when studying compounds that bind to intrinsically disordered proteins, but I believe that this is a general problem in drug design, including compounds that bind to folded proteins.

Josep Rizo

Reviewer #3 (Remarks to the Author):

This paper examines the binding of two proposed small molecule drugs to the disordered transactivation domain of the androgen receptor. This is of interest because of the (so far unrealized) potential for targeting intrinsically disordered regions/proteins with drugs. Both drugs are found to result in a collapse of the protein relative to the apo state, with formation of helical structure, these effects being more pronounced in the presence of the EPI-7170 ligand than the EPI-002. Residues in the proteins responsible for the binding are identified. Because of the novelty of this strategy and the new mechanistic information provided by the simulations, I anticipate that this paper will be of broad interest.

I suggest the authors address the following comments in revision:

1. The authors say the simulations are "relatively well converged", which is a little bit meaningless. Of course the simulations are not converged (if they were, all the curves in e.g. Fig. S4 would lie on top of each other), but that is to be expected. What matters is whether the properties that are being averaged have converged -- this can be assessed by plotting windowed averages as a function of simulation time to show that these have reached a stable value and aren't still systematically shifting.
2. Is it correct that the simulations of the R2-R3 region are being compared with experiments of a larger peptide including helix R1? Although significant Csp's for R1 with the drug were not seen for EPI-002, that doesn't rule out they could exist for EPI-7170. This should be pointed out / discussed.
3. With a mM affinity, I am not sure it makes sense to talk about bound/unbound conformations of the protein, as the association/dissociation rates are likely to be on a similar or even faster time scale than conformational rearrangements (certainly than formation/breaking of the helical structure).
4. In figure 2C, should the diagonal not show the same data as Fig. 2A? The maximum of the color bar in 2C is at 0.25, but the data in 2A go up to 0.4. It's possible that 2C is referring only to the "bound ensemble" as labelled and not the whole simulation, but then one would expect the contact probabilities to be higher and not lower.

Reviewer #4 (Remarks to the Author):

This is a manuscript that uses all-atom molecular dynamics computer simulations to elucidate atomically detailed binding mechanisms of EPI-002 and a more potent analogue, EPI-7170, to the known binding site of EPI-002 in Tau-5 of the amino-terminal domain (NTD) of the androgen receptor (AR) that was revealed previously by NMR. Two of the three regions (R2 and R3) of AR-NTD known to interact with EPI-002 are utilized while omitting R1, in spite of published data that all three regions (residues 354-448) must simultaneously be present for EPI-002 to bind (de Mol et al 2016). EPI-7170 has been shown by others to be more potent than EPI-002 in biological assays for blocking AR transcriptional activity, but it has not been shown that EPI-7170 binds to AR. EPI-7170 differs from EPI-002 by having two chlorine atoms on one of the phenol rings and in place of an alcohol has a MeO₂SHN group. These studies here are restricted to all-atom explicit solvent MD simulation of Tau-5R2-R3 and support published biological data in terms of EPI-7170 being more potent than EPI-002. The authors suggest specific amino acid residues within R2 and R3 of the AR-NTD and mechanisms that may underlie this difference in potency. No biological data is provided to support conclusions of the importance of these specific residues identified in the potential binding mechanisms of EPI-7170 to AR-NTD.

Concerns

- 1) Biological support of the conclusion that the aromatic residues W397 and W433 are important in EPI-7170 binding to the AR are lacking, or that they impact potency. It is not been proven that EPI-7170 binds AR.
- 2) These studies would benefit from inclusion of the more structurally related compounds such as BADGE-2H₂O and iodo-EPI-002. BADGE-2H₂O has no biological activity (Andersen et al 2010) and is only one atom different from EPI-002; whereas iodo-EPI-002 compound has been proven to bind to the AR-NTD, is extremely potent, and also is only one atom different from EPI-002 (Imamura et al 2016). The EPI-7170 compound has not been proven to bind to the AR and has a substantially differing chemical structure from EPI-002.
- 3) Restriction of the studies to only Tau-5R2-R3 raises uncertainty about whether similar results would be obtained with these compounds with an unrelated protein sequence. Previous NMR work published with EPI-002 showed less binding to Tau-1 containing more structured regions. As is there is no control for the protein sequence in these studies.
- 4) AR-NTD is rich in cysteine residues which play an important role in protein structure through disulfide bridges. Some discussion about this in relation to the presented work would be helpful.
- 5) The binding site contains residues that are known to be post-translationally modified (for example pS424). Some discussion on such modifications and limitations of the modeling approach would be beneficial.

6) References about covalent binding need to be corrected in the last paragraph of the Discussion that incorrectly cite ref 37, whereas references 38 and 39 address this point.

We found the reviewer's suggestions and comments extremely helpful. We have performed several additional calculations and analyses which have been added to the revised manuscript. A detailed response to each reviewer's comments is included below.

Reviewer #1

EPI-002 and EPI-7170 are two compounds that were developed against the castration-resistant prostate cancer. These two compounds were shown to mainly bind to the R2 and R3 in the Tau-5 in the AR-NTD. Zhu et al. used molecular dynamics simulations to analyze the binding behavior of two compounds with a 56-residue segment containing R2 and R3. However, the simulated conformations contain obviously larger helical content than previous NMR results.

We thank the reviewer for raising this important point. We did not discuss the quantitative agreement between our simulations and previous NMR results and the relative sources of error in structure-based NMR chemical shift prediction and NMR chemical shift based secondary structure prediction algorithms in sufficient detail in the previous version of our manuscript. We have added the following revised text to address this comment on pages 9 and 10 of the revised manuscript.

“We validated the accuracy of our MD simulation of Tau-5_{R2_R3} by calculating NMR chemical shifts with SPARTA[±]⁵⁷ and comparing them to previously reported experimental values²⁴ (Table 1, Fig. 1). The overall agreement between calculated and experimental backbone chemical shifts is excellent and is on-par with the most accurate chemical shift predictions reported in force field benchmarks of simulations of IDPs^{49,58}. Specifically, we note that the RMSD between calculated and experimental NMR chemical shifts observed for this simulation ($C\alpha$ RMSD = 0.47ppm) is among the lowest values reported for unbiased MD simulations of disordered proteins in an extensive benchmark of long time-scale MD simulations of 9 IDPs run with 7 state-of-the-art forcefields, and is lower than the RMSD between calculated experimental shifts for several experimentally derived IDP ensembles obtained using NMR chemical shifts as restraints⁴⁹. The predicted $C\alpha$ shifts are within the estimated 0.92ppm SPARTA[±] $C\alpha$ shift prediction error for all residues with the exception of slightly larger deviations in residues ⁴³⁴WH⁴³⁵ (Fig. 1C). In addition to a direct comparison with NMR chemical shifts, we also observed that the simulated helical propensities of residues ⁴³¹SSWHTLF⁴³⁷ in the R3 region, calculated using the DSSP algorithm, are somewhat overestimated relative to those estimated from NMR chemical shifts using the secondary structure propensity prediction algorithms Delta2D and ncSCP^{59,60} (Fig. 1A). We note that chemical shift based secondary structure propensity prediction algorithms such as Delta2D and ncSCP are subject to systematic and sequence specific errors and show larger deviations with helical propensity estimates from circular dichroism (CD) for secondary structure elements with a population <30%⁵⁹.”

Additionally, we have expanded our discussion of the results of maximum-entropy reweighting methods we have used to rigorously quantify deviations from experimental NMR chemical shifts.

“To rigorously quantify the error in the simulated helical propensity directly against experimental data, we utilized the maximum-entropy reweighting algorithm of Cesari et al.^{61,62} (See Methods) to reweight our unbiased 300K ensemble using $C\alpha$ chemical shifts as restraints (Table 1, Fig. 1). We note that while only $C\alpha$ chemical shift predictions were used as restraints in the reweighting procedure, we observed improvements in the prediction accuracy of the remaining backbone shifts in the reweighted ensemble, suggesting that the resulting ensemble is not overfit to the $C\alpha$ chemical shift data. We found that optimal agreement with experimental shifts was obtained by reducing the average helical propensity of residues ⁴³¹SSWHTLF⁴³⁷ calculated using the DSSP algorithm from 34% in the unbiased ensemble to 25%

in reweighted ensemble, suggesting an overstabilization of helical conformations in this region by 0.26kcal/mol (or ~0.04 kcal/mol per residue in this 7-residue segment) in the unbiased trajectory. This suggests that improved agreement with NMR chemical shifts could potentially be obtained in unbiased simulations by applying a residue specific force field torsion correction to these residues⁶¹ however, free energy differences this small may be difficult to resolve beyond the statistical uncertainty of conformational sampling in simulations of IDPs of this size. We note that the C α -reweighted maximum entropy ensemble still possesses more helical content in the R3 region than predicted by the NMR chemical shift based algorithms Delta2D and ncSCP, though this level of discrepancy is potentially within the error of these secondary structure propensity prediction methods (Fig. 1A). Considering that both chemical shift predictions from SPARTA \pm ⁵⁷ and NMR chemical shift based secondary structure prediction algorithms^{57,59,60} are subject to prediction errors, we consider the relatively modest deviation from experimental chemical shifts and predicted secondary structure propensities to be acceptable.

We note that while unbiased Tau-5_{R2_R3} simulations run with the a99SB-*disp* force field may slightly overestimate the stability of the R3 helix, the simulated helical conformations in these regions are still only marginally stable with an average population of 34% in the 300K replica of our REST2 simulation. As these simulations sample both helical and non-helical states, we expect that unbiased simulations will appreciably sample ligand binding modes with both helical and non-helical states of the R3 region. We therefore expect that an unbiased simulation should be capable of resolving increases or decreases in the stability of helical conformations in the presence of ligands, even if the absolute populations of helical conformations in simulated bound and unbound states are elevated relative to solution experiments. It is possible however, that a slight overstabilization of helical conformations in the R3 region in the apo ensemble of Tau-5_{R2_R3} may make an increase in the stability of helical conformations in this region in the presence of ligands more difficult to detect, and simulations in the presence of ligands may underestimate increases in helical propensity of the R3 region in the presence of ligands.

Finally we note that in our experience analyzing the conformational ensembles of IDPs, direct comparisons between structure-based NMR chemical calculations with experimental values are a more robust and reliable measure of the accuracy of an IDP conformational ensemble relative to indirect secondary structure propensity prediction algorithms from backbone chemical shifts as quantitatively assessed by agreement with cross-validating solution data (Robustelli et al. JCTC, 16 (4), 2020, Robustelli et al. PNAS, 115 (21), 2018, Robustelli et al. JCTC, 9 (11), 2013).

Although the simulations showed that EPI-7170 might bind more stronger than EPI-002, both of them are simulated to have KD values in mM range, which are very weak.

The reviewer raises an important point about making direct comparisons between simulated K_D values from molecular dynamics computer simulations, spectroscopically determined K_D values, and biological potencies from cellular assays and animal models that was not sufficiently discussed in the initial manuscript. We have added additional discussion of this point on pages 12-14 of the revised manuscript:

“We note that the absolute values of intermolecular contact probabilities and the corresponding K_D values will be sensitive to the distance thresholds used to define intermolecular contacts and that a purely distance-based definition of “bound” frames will also count transient collisions between ligands and proteins among bound conformations. This is not inherently problematic, however, as experimental spectroscopic methods used to detect ligand binding such as NMR spectroscopy may also be sensitive to such transient collisions. We note that different

spectroscopic methods used to measure experimental binding affinities of ligands to disordered proteins will be sensitive to different features of ligand binding modes to varying extents. In particular, an atomic resolution spectroscopy such as NMR will detect interactions with each residue independently and the strength of the signal will depend on the identity of the chemical moieties that are brought into proximity upon binding. Measurements from surface plasmon resonance (SPR), bilayer interferometry, isothermal calorimetry (ITC), or fluorescence anisotropy may be more globally sensitive to binding and report on the total number of molecules in solution that contain any contacts with ligands. In several instances, small molecule ligands that appear to bind IDPs with relatively weak mM affinities as assessed by residue-level NMR chemical shift perturbations appear to bind substantially tighter, in the low μM affinity range, using other spectroscopic techniques such as SPR and bilayer interferometry^{27-28,30}. Understanding these relationships well enough to quantitatively compare simulated and spectroscopically measured binding affinities will likely require extensive experimental and computational benchmarking. Simulated K_D values are therefore most meaningfully compared to simulated K_D values calculated with the same distance thresholds and may not be directly comparable to experimental K_D values measured from biophysical experiments and spectroscopic techniques beyond ranking the relative affinities of ligands in a series.

We and others³² have found that the distance threshold used to define intermolecular contacts has little effect on the ratios of K_D values and contact probabilities calculated for a series of small molecule ligands in separate binding simulations. The distance threshold employed in the definition of bound conformations can therefore be thought of as a scaling factor for simulated K_D values and the optimal distance threshold for comparisons to experimental K_D values is likely to vary based on the experimental techniques used. In single replica unbiased MD simulations, one can utilize the residence time of ligand binding events to differentiate transient encounters from more stable binding events³², but applying residence time-based thresholds for bound conformations is less straightforward when using enhanced sampling techniques such as REST2⁵³.

We further note that EPI-002, which produces small NMR CSPs consistent with a mM *in vitro* binding affinity and has a simulated mM K_D value, has been shown to have of $\sim 10\mu\text{M}$ IC_{50} value in prostate-specific antigen luciferase reporter assays for inhibition of endogenous AR transcriptional activity in cellular assays as well as clear anti-tumor activity in mouse models³⁷. We therefore caution that the magnitudes of experimental K_D values from biophysical assays and simulated K_D values from molecular dynamics computer simulations are not clear predictors of biological activity of IDP ligands. There is not clear evidence to suggest that nanomolar affinity binding is required for IDP ligands to exhibit biological activity. This likely results from the fact that IDPs often have central roles in cellular signaling networks that can involve multivalent low affinity interactions with a large number of physiological interaction partners, as well as the importance of both kinetics and thermodynamics in the formation of higher order molecular species involved in protein misfolding and biomolecular condensate formation in cellular contexts^{8,9,20-23}.

Compared to what is known before, this study does not offer much new insight, as it is already known that the compounds mostly bind to the R2 and R3 range and increase helical content.

While previously reported NMR measurements localized the largest NMR chemical shift perturbations of EPI-002 to the R1, R2, & R3 helices of the Tau-5 region of AR, these measurements in isolation provide no atomistic information about the binding mechanisms of EPI compounds that can be translated into medicinal chemistry insights. For example, based on previous NMR results, it is not possible to make rational, structure-based, predictions of how modifications to the chemical structure of EPI-002 or mutations of Tau-5 would influence the specificity and affinity of this compound.

In this work we report the elucidation of atomic resolution binding mechanisms that can be experimentally probed and refined with Tau-5 mutants and newly designed small molecules. To our knowledge, this is the first such mechanism reported for a clinical drug candidate known to target an IDP. We believe this work represents an important milestone in the emerging field of intrinsically disordered protein drug design and that this manuscript will be of broad interest to the intrinsically disordered protein and computational drug design communities.

The other problem is that EPI-002 was known to form covalent adduct with the protein (ref. 24), which was not considered in the simulations.

We thank the reviewer for raising the point of the covalent reactivity, which was only briefly mentioned in the discussion section of the previous version of the manuscript. We have performed additional analyses and substantially expanded our discussion of covalent reactivity in the revised manuscript to more clearly indicate our hypothesis that the non-covalent binding affinity of EPI ligands directs site-specific covalent attachment to specific cysteines in Tau-5, and that tighter non-covalent binding affinities of EPI ligands will increase the covalent reactivity of these ligands at specific cysteines. We have added two new sections to the manuscript: “Non-covalent binding of EPI-7170 & EPI-002 increases the proximity of their weakly reactive chlorohydrin groups to the nucleophilic thiol of CYS404 in collapsed helical states of Tau-5_{R2_R3}” and “Binding Mode Comparisons of EPI-002 and EPI-7170 with Iodo-EPI-002 and BADGE:2H₂O”. These sections include a new analysis examining the distance distribution between the reactive chlorohydrin groups of EPI-002 and EPI-7170 with the thiol group of CYS404 in ligand binding simulations, as well the analysis of a new 67 μ s REST2 simulation of the non-covalently reactive analog of EPI-002, BADGE:2H₂O, which has a diol group in place of a chlorohydrin group exploring these hypotheses. The text of these sections is included below. Additionally, we have added 4 new

figures to the Supplementary information (Fig S31-S34) and an additional table (Table S1) in support of this hypothesis.

“Non-covalent binding of EPI-7170 & EPI-002 increases the proximity of their weakly reactive chlorohydrin groups to the nucleophilic thiol of CYS404 in collapsed helical states of Tau-5_{R2_R3}

EPI-002 and EPI-7170 both possess a chlorohydrin group (Fig. 2). The chlorohydrin group of EPI-002 has been shown to be weakly reactive with cysteine residues in the AR-NTD, and it is hypothesized that covalent attachment of EPI compounds to Tau-5 may be required for its biological activity^{14,37,38}. It has previously been proposed that fast reversible non-covalent interactions of EPI compounds to different regions of the AR-NTD may drive covalent attachment to specific cysteines in AR³⁸. Based on previously reported NMR experiments on the AR-NTD that showed the largest backbone amide NMR CSPs in the R2 and R3 regions²⁴ in EPI-002 titrations, we hypothesize that non-covalent binding of EPI ligands may direct covalent attachment of these ligands to CYS404. CYS404 is found in the middle of the transiently helical R2 region (³⁹⁷WAAAAAQCRYG⁴⁰⁷) before the helix breaking GLY407 residue. If covalent reactivity at CYS404 is important for the biological activity of EPI ligands, we suspect that EPI ligands with non-covalent binding modes that more effectively localize their weakly reactive chlorohydrin groups to the nucleophilic thiol of CYS404 will be more potent AR inhibitors *in vivo*. We compare the distance distributions between the chlorohydrin chlorine atoms of EPI-7170 and EPI-002 to the sulfur atom of CYS404 of Tau-5_{R2_R3} in our ligand binding simulations in Fig. S31. We find that the higher non-covalent affinity of EPI-7170 relative to EPI-002 dramatically increases the proximity of its chlorohydrin group to CYS404 relative to the distance distribution observed between the chlorohydrin group of EPI-002 and CYS404. We find that the chlorohydrin chlorine atom of EPI-7170 is within 10.0Å of the CYS404 thiol sulfur atom in 23.0% of simulation frames while the chlorine atom of EPI-002 is within 10.0Å of the CYS404 thiol sulfur atom in 9.6% of simulation frames. We note that in addition to increasing the proximity of the EPI ligand chlorohydrin groups to CYS404, the compact nature of the ligand bound ensembles may further facilitate covalent attachment by sequestering the reaction from solvent water molecules. The conformational properties of covalent adducts of EPI compounds bound to Tau-5 will be the subject of future investigations.

Binding Mode Comparisons of EPI-002 and EPI-7170 with Iodo-EPI-002 and BADGE:2H₂O

To obtain additional insight into the inhibition mechanisms of EPI-002 and EPI-7170 we also conducted REST2 MD simulations of Tau-5_{R2_R3} in the presence of two additional ligands: Bisphenol A Diglycidic Ether:2H₂O (BADGE:2H₂O), and Iodo-EPI-002 (Figure S32). BADGE:2H₂O has an identical structure to EPI-002 aside from the replacement of the EPI-002 chlorohydrin group with a diol. This substitution eliminates the covalent reactivity of the compound, and cellular assays have shown that BADGE:2H₂O does not inhibit AR transcriptional activity^{37,38}. Iodo-EPI-002 differs from EPI-002 only by the addition of a single iodine atom on the bisphenol A phenyl ring closest to the chlorohydrin group and has been shown to be an ~10x more potent AR inhibitor in cellular AR transcriptional activity inhibition assays³⁹. A REST2

simulation of BADGE:2H₂O and Tau-5_{R2_R3} was run for 4.2μs/replica, for an aggregate simulation time of 56μs, and a REST2 simulation of Iodo-EPI-002 and Tau-5_{R2_R3} was run for 5.0μs/replica, for an aggregate simulation time of 80μs. The simulated properties of Tau-5_{R2_R3} observed in these simulations are compared to the apo Tau-5_{R2_R3} simulations in Table S1 and Fig. S32-S33. Simulation and convergence analyses for these simulations are reported in Fig S32-S33.

We observe that BADGE:2H₂O has a simulated K_D of 5.61 ± 0.29 mM, which is slightly larger than, but within statistical uncertainty estimates of, the simulated K_D values of EPI-002 (5.24 ± 0.43 mM). We observe that most simulated properties are within statistical uncertainty estimates of the simulated properties of EPI-002 (Table S1, Figure S32-S33) and that the intermolecular interactions between Tau-5_{R2_R3} and each ligand observed in the bound ensembles are extremely similar (Figure S34). The similar simulated binding affinities and properties of the bound ensembles of EPI-002 and BADGE:2H₂O are consistent with the hypothesis that differences in their biological potency may strictly result from a lack of covalent reactivity, rather than from differences in their non-covalent binding affinity or non-covalent binding mechanisms. These results are consistent with the hypothesis that covalent attachment is essential for the biological activity of EPI AR-NTD inhibitors.

Iodo-EPI-002 has been shown to be substantially more potent than EPI-002 in cellular AR inhibition assays³⁹. Iodo-EPI-002 was shown to have IC₅₀ values of ~1μM in prostate-specific antigen luciferase reporter assays for inhibition of endogenous AR transcriptional activity³⁹ compared to IC₅₀ values of ~10μM for EPI-002 in similar assays³⁷⁻³⁸. EPI-7170 was also found to have an IC₅₀ value of ~1μM in cellular prostate-specific antigen luciferase reporter assays⁴⁶. We found that the simulated K_D of Iodo-EPI-002 (1.89 ± 0.13 mM) is extremely similar to the simulated K_D of EPI-7170 (1.92 ± 0.15 mM). Interestingly, we observe that the Iodo-EPI-002:Tau-5_{R2_R3} bound ensemble has a larger fraction helix ($38.4 \pm 1.7\%$) and helical globule population ($79.2 \pm 3.3\%$) compared the EPI-7170:Tau-5_{R2_R3} bound ensemble (fraction helix of $32.8 \pm 0.5\%$ and helical globule population of $61.1 \pm 4.5\%$). We observe that the populations of intermolecular interactions in the ligand Iodo-EPI-002:Tau-5_{R2_R3} bound ensembles are very similar to those observed in EPI-7170:Tau-5_{R2_R3} bound ensemble (Figure S34). These simulation results suggest that Iodo-EPI-002 and EPI-7170 bind Tau-5 with similar binding mechanisms and similar affinities and are consistent with the similar potencies observed in cellular AR transcriptional inhibition assays.

We also have added the following to the discussion session on page 25 of the revised manuscript

“The chlorohydrin group of EPI-002 has been found to be weakly covalently reactive with Tau-5, and it is hypothesized that covalent attachment of EPI compounds to Tau-5 is required for their biological activity^{14,37-39}. We have simulated Tau-5_{R2_R3} binding the compound BADGE:2H₂O, which replaces the chlorohydrin group EPI-002 with an unreactive diol. We observe that EPI-002 and BADGE:2H₂O have very similar simulated binding affinities, and that the EPI-002:Tau-5_{R2_R3} and BADGE:2H₂O:Tau-5_{R2_R3} bound ensembles are extremely similar, in agreement with the hypothesis that covalent reactivity is essential for AR-NTD inhibition in cellular contexts. In our simulations, we observe that reversible non-covalent binding of EPI-002 and EPI-7170 at the

dynamic interface between the R2 and R3 regions of Tau-5 increases the proximity of their weakly reactive chlorohydrin groups to the thiol group of cysteine 404, and that the higher affinity binding of EPI-7170 more effectively localizes its chlorohydrin group to the cysteine 404 thiol. We hypothesize that the non-covalent affinity of EPI compounds to this region may drive preferential covalent attachment to this cysteine residue relative to other cysteine residues in the AR-NTD, which may further stabilize compact helical conformations of Tau-5 and inhibit the ⁴³³WHTLF⁴³⁷ segment of the R3 helix from binding the RAP74 domain of the general transcription regulator TFIIF^{47,48,51,52}. In this scenario, we expect that compounds with higher non-covalent binding affinity to Tau-5_{R2_R3}, such as EPI-7170 and Iodo-EPI-002, will be more covalently reactive in cellular contexts. The conformational properties of covalent adducts of EPI compounds bound to Tau-5 will be the subject of future investigations.”

The reviewer is correct that we have not included simulations of the covalent adducts in this work. The Robustelli and Salvatella laboratories are currently conducting computational and experimental investigations of the conformational properties of covalent adducts of EPI ligands. Simulations of the covalent adducts, which require careful parametrization and validation of the covalent adduct force field parameters are currently underway. These simulations will take several months to sufficiently converge, and may require additional tuning and optimization based on the results of ongoing experimental NMR studies. We believe that these results are outside the scope of the current investigation and will be the subject of future work.

Reviewer #2

This paper describes an interesting study designed to elucidate how small compounds bind to an intrinsically disordered fragment of the androgen receptor (AR) using molecular dynamics (MD) simulations. This is an area of the utmost interest for research on prostate cancer because development of castration resistant prostate cancer is known to depend critically on the activity of the N-terminal transactivation domain of AR (AR NTD), which is intrinsically disordered. Small compounds that binding to this domain are currently in clinical trials to treat prostate cancer. Understanding how the compounds bind to the domain is therefore crucial to optimize the compounds and increase their potency, but defining the binding mode is strongly hindered by the weak affinity of the compounds and by the intrinsically disordered nature of the AR NTD. Although NMR spectroscopy provided some insights into the regions of the AR NTD that bind to some of the existing compounds, such as EPI-002, it is extremely difficult to determine the binding modes in atomic detail by any currently available experimental method.

Zhu et al. describe a commendable computational approach to shed light into how EPI-002 and a related more potent compound (EPI-7170) bind to the region of the AR NTD that was found to bind to EPI-002 by NMR spectroscopy. The MD simulations that they describe suggest that the compounds bind dynamically (i.e. in multiple modes) to the interface between two helices through networks of aromatic ring stacking interactions and hydrogen bonds. In all honesty, I am not sure to what extent the conformations and binding modes visited during the simulations represent the conformational ensembles and binding modes existing in solution. However, the observed binding modes make sense from a chemical point of view and do involve the residues that exhibited larges

chemical shift changes upon addition of EPI-002 in previous NMR studies. Moreover, the authors make a strong effort to correlate the observed NMR chemical shifts with those predicted for the ensembles of conformations observed during the simulations. Hence, I believe that this is currently the best we can do to understand the binding modes. Overall, this study will be of strong interest to researchers on prostate cancer and also in general to those interested in binding of organic compounds to intrinsically disordered regions of proteins.

We thank the reviewer for these supportive comments.

I do have a few minor concerns that should be addressed before publication.

1. The authors do a reasonable job at not overselling their results and consider the limitations of their approach, but I would still encourage them to be careful with wording and tone down strong conclusions. As an example, I would prefer that the authors say that the NMR data ‘are consistent with’ or ‘support the validity’ of the MD simulation results rather than saying that the NMR data ‘validate’ the results.

We thank the reviewer for this suggestion, and we have amended all instances in the text accordingly:

Abstract, Page 2:

“Our simulations, which are highly consistent with measurements from NMR spectroscopy”

Discussion, Page 23:

“In this investigation, we have leveraged recent advances in molecular simulation force fields and enhanced sampling methods to provide atomic resolution binding mechanism models of EPI-002 and EPI-7170 that are highly consistent with previously reported NMR measurements.”

Page 24:

“In this investigation, we have assessed the accuracy our simulations of apo Tau-5_{R2_R3} and EPI-002 Tau-5_{R2_R3} binding by comparisons with previously measured NMR data. The consistency between our simulations and NMR data suggests that the simulation methods and force fields utilized are well suited to describe the conformational properties of Tau-5_{R2_R3} and the interactions between Tau-5_{R2_R3} and EPI compounds with a common bisphenol-A scaffold. “

2. Since one of the main interests of this study is the atomic detail information provided by the simulations on the interactions between the small compounds and the AR NTD, they should provide better visualization of these details. The structures shown in fig. 2D should be much larger (please protect people’s eyes and do not assume that they can just expand the figure on a computer screen; figures should be readily evaluated on a printed copy). Moreover, they should show a representative gallery of the diverse binding modes observed in main figures (not in supplementary materials). The figures should allow the reader to observe how similar or different are the most populated binding modes observed for a given compound, and to observe the key interactions. Note that movies are useful, but cannot replace static pictures that can be studied calmly.

We thank the reviewer for this excellent suggestion. We have now added additional analyses and figures to better capture the conformational diversity of apo and bound ensembles of Tau-5_{R2_R3}. (Figure S7, Figure S20, Figure S21). These figures display the free energy surface of the Tau-5_{R2_R3} apo and bound ensembles as a function of the alpha helical order parameter $S\alpha$ and the number of contacts between aromatic residues (where the ligands are all considered as an additional aromatic residue in the bound ensembles). These figures illustrate the diversity of the sizes of aromatic cores in the conformational ensembles of Tau-5_{R2_R3}, and provide snapshots of structures within free energy basins with similar sized aromatic cores, which illustrate their conformational diversity. Due to the size of these figures, and the need for larger images to be able to distinguish the conformational features of the members of these ensembles, we do not believe we could add them to the main text with sufficient resolution.

We note that these conformations are extremely heterogenous, so it would not be possible to capture the full conformational variability of these states without images of 100s of structures, and the development of rigorous clustering algorithms to present IDP ensembles as a tractable number of visually interpretable representative states remains an ongoing research effort in our laboratory and others.

For these reasons, we have also deposited a 1000 structure of the NMR chemical shift reweighted conformational ensemble of apo Tau-5_{R2_R3} in the protein ensemble database (<https://proteinensemble.org/PED00206>) where it can be downloaded and further visualized/analyzed. Unfortunately, we were not permitted to deposit the bound ensembles, as the protein ensemble database does not host unbiased MD trajectories. These ensembles (containing 10,000s of bound frames) along with the entire apo ensemble, are available for download in the code repository accompanying this manuscript, (https://github.com/paulrobustelli/AR_ligand_binding) and will be uploaded to zenodo upon publication of the final manuscript.

3. The criterion used to compute population of bound states, P_b , seems to be conceptually flawed, as binding is assigned simply from the proximity of two atoms from different molecules. If I understand correctly, this means that random encounters where two atoms come close in a transient manner, without staying close because of interactions between them, are considered as bound. Since the same criterion was used for the two compounds, the relative affinities measured for the two compounds may still be fine, but this limitation in the limitation should be pointed out. Hopefully, the field will revise this methodology in the future.

We thank the reviewer for raising this important point about the subtleties of portioning trajectories into bound and unbound frames and calculating K_D values for dynamic binding of small molecules interacting with IDPs. As stated in our response to Reviewer 1, that raised a related point, we have now added a substantial discussion of this issue on pages 12-14 of the revised manuscript:

“We note that the absolute values of intermolecular contact probabilities and the corresponding K_D values will be sensitive to the distance thresholds used to define intermolecular contacts and that a purely distance-based definition of “bound” frames will also count transient collisions

between ligands and proteins among bound conformations. This is not inherently problematic however, as experimental spectroscopic methods used to detect ligand binding such as NMR spectroscopy may also be sensitive to such transient collisions. We note that different spectroscopic methods used to measure experimental binding affinities of ligands to disordered proteins will be sensitive to different features of ligand binding modes to varying extents. In particular, an atomic resolution spectroscopy such as NMR will detect interactions with each residue independently and the strength of the signal will depend on the identity of the chemical moieties that are brought into proximity upon binding. Measurements from surface plasmon resonance (SPR), bilayer interferometry, isothermal calorimetry (ITC), or fluorescence anisotropy may be more globally sensitive to binding and report on the total number of molecules in solution that contain any contacts with ligands. In several instances, small molecule ligands that appear to bind IDPs with relatively weak mM affinities as assessed by residue-level NMR chemical shift perturbations appear to bind substantially tighter, in the low μM affinity range, using other spectroscopic techniques such as SPR and bilayer interferometry^{27-28,30}. Understanding these relationships well enough to quantitatively compare simulated and spectroscopically measured binding affinities will likely require extensive experimental and computational benchmarking. Simulated K_D values are therefore most meaningfully compared to simulated K_D values calculated with the same distance thresholds and may not be directly comparable to experimental K_D values measured from biophysical experiments and spectroscopic techniques beyond ranking the relative affinities of ligands in a series.

We and others³² have found that the distance threshold used to define intermolecular contacts has little effect on the ratios of K_D values and contact probabilities calculated for a series of small molecule ligands in separate binding simulations. The distance threshold employed in the definition of bound conformations can therefore be thought of as a scaling factor for simulated K_D values and the optimal distance threshold for comparisons to experimental K_D values is likely to vary based on the experimental techniques used. In single replica unbiased MD simulations, one can utilize the residence time of ligand binding events to differentiate transient encounters from more stable binding events³², but applying residence time based thresholds for bound conformations is less straightforward when using enhanced sampling techniques such as REST2⁵³. In a previous study of small molecules binding a 20-residue fragment of α -synuclein that utilized unbiased MD simulations and the same force fields employed in this investigation³², a broad distribution of residence times, from 1ns-2 μ s was observed. Based on the higher simulated binding affinities of the compounds studied here, we speculate that the distribution of residence times observed in unbiased MD simulations of EPI-002 and EPI-7170 binding Tau-5_{R2_R3} would be shifted to longer residence times.

We further note that EPI-002, which produces small NMR CSPs consistent with a mM *in vitro* binding affinity and has a simulated mM K_D value, has been shown to have of $\sim 10\ \mu\text{M}$ IC_{50} value in prostate-specific antigen luciferase reporter assays for inhibition of endogenous AR transcriptional activity in cellular assays as well as clear anti-tumor activity in mouse models³⁷. We therefore caution that the magnitudes of experimental K_D values from biophysical assays and simulated K_D values from molecular dynamics computer simulations are not clear predictors of biological activity of IDP ligands. There is not clear evidence to suggest that nanomolar affinity binding is required for IDP ligands to exhibit biological activity. This likely results from the fact

that IDPs often have central roles in cellular signaling networks that can involve multivalent low affinity interactions with a large number of physiological interaction partners, as well as the importance of both kinetics and thermodynamics in the formation of higher order molecular species involved in protein misfolding and biomolecular condensate formation in cellular contexts 8,9,20-23 .”

4. In the discussion the authors state that solubility may be a ubiquitous problem when studying compounds that bind to intrinsically disordered proteins, but I believe that this is a general problem in drug design, including compounds bind to folded problems.

Josep Rizo

This is a valuable point and we have added the following additional text to the discussion section on page 24:

“Obtaining sufficiently soluble small molecules drugs is often a challenge in drug discovery campaigns targeting folded proteins, where active ligands can have very tight picomolar-nanomolar affinities. As many small molecule disordered proteins ligands with biological activity have been found to have relatively low μM - mM affinities in biophysical assays, obtaining sufficient solubility for detailed biophysical characterization of small molecule ligands may be particularly challenging in drug discovery campaigns for disordered targets.”

Josep Rizo

Reviewer #3 (Remarks to the Author):

This paper examines the binding of two proposed small molecule drugs to the disordered transactivation domain of the androgen receptor. This is of interest because of the (so far unrealized) potential for targetting intrinsically disordered regions/proteins with drugs. Both drugs are found to result in a collapse of the protein relative to the apo state, with formation of helical structure, these effects being more pronounced in the presence of the EPI-7170 ligand than the EPI-002. Residues in the proteins responsible for the binding are identified. Because of the novelty of this strategy and the new mechanistic information provided by the simulations, I anticipate that this paper will be of broad interest.

I suggest the authors address the following comments in revision:

1. The authors say the simulations are "relatively well converged", which is a little bit meaningless. Of course the simulations are not converged (if they were, all the curves in e.g. Fig. S4 would lie on top of each other), but that is to be expected. What matters is whether the properties that are being averaged have converged -- this can be assessed by plotting windowed averages as a function of simulation time to show that these have reached a stable value and aren't still systematically shifting.

We thank the review for this suggestion. We have added convergence plots for the relative conformational properties of Tau-5_{R2_R3} discussed in this manuscript ($S\alpha$, R_g , Helical Globule Population, and K_D) in Figures S22, S26, S27, and S32.

2. Is it correct that the simulations of the R2-R3 region are being compared with experiments of a larger peptide including helix R1? Although significant Csp's for R1 with the drug were not seen for EPI-002, that doesn't rule out they could exist for EPI-7170. This should be pointed out / discussed.

The reviewer is correct that we are comparing results to NMR chemical shift perturbations in EPI-002 titrations measured in the context of a full length Tau-5 construct that contains the R1 helix and this point warrants additional discussion in the manuscript. We have now added additional discussion of this point in the discussion section on page 27 of the revised manuscript:

“The simulations conducted in this study were carried on a fragment of the Tau-5 region of AR containing the residues previously observed to have the highest propensity to interact with EPI-002 to obtain a computationally tractable system and enable statistically meaningful comparisons of the binding modes of EPI-002 and EPI-7170. We note that the R1 helix of Tau-5 was also found to interact with EPI-002 by NMR in the context of the full Tau-5 region but that a peptide containing only the R1 domain was not found to interact with EPI-002 at similar concentrations. Based on the sequence composition of R1, in particular the presence of 6 aromatic residues in a 31-residue domain and the helical propensity observed by NMR, we speculate that EPI-002 and EPI-7170 likely interact with the with R1 region with similar bindings mechanisms to those observed in this investigation, and that EPI-7170 transiently binds to R1 with a higher affinity than EPI-002. We hypothesize that EPI compounds stabilize transient long-range intramolecular between the R1 and R2 regions, R1 and R3 regions, and simultaneous interactions between the R1, R2 and R3 regions with similar binding modes to those observed between the R2 and R3 regions in this investigation. Due to the presence of the relatively rigid polyproline linker between R1 and R2, we suspect that interactions involving the R1 helix are less important for conferring the affinity of EPI compounds to Tau-5 than the interactions between R2 and R3 described here. Studying the interactions of EPI compounds with larger constructs of Tau-5 that contain the R1, R2 and R3 regions will require substantially longer simulations than those reported here, and in practice will likely require the optimization of additional enhanced sampling algorithms for this system.”

3. With a mM affinity, I am not sure it makes sense to talk about bound/unbound conformations of the protein, as the association/dissociation rates are likely to be on a similar or even faster time scale than conformational rearrangements (certainly than formation/breaking of the helical structure).

The reviewer raises an important point about distinguishing bound and unbound conformations in dynamic binding events and time scales of association and dissociation for these binding events. As stated in our responses to related points raised by Reviewers 1 and 2 we have included an detailed discussion of these points in the results section of the revised manuscript on pages 12-13:

“We note that the absolute values of intermolecular contact probabilities and the corresponding K_D values will be sensitive to the distance thresholds used to define intermolecular contacts and that a purely distance-based definition of “bound” frames will also count transient collisions between ligands and proteins among bound conformations. This is not inherently problematic however, as experimental spectroscopic methods used to detect ligand binding such as NMR spectroscopy may also be sensitive to such transient collisions. We note that different spectroscopic methods used to measure experimental binding affinities of ligands to disordered proteins will be sensitive to different features of ligand binding modes to varying extents. In particular, an atomic resolution spectroscopy such as NMR will detect interactions with each residue independently and the strength of the signal will depend on the identity of the chemical moieties that are brought into proximity upon binding. Measurements from surface plasmon resonance (SPR), bilayer interferometry, isothermal calorimetry (ITC), or fluorescence anisotropy may be more globally sensitive to binding and report on the total number of molecules in solution that contain any contacts with ligands. In several instances, small molecule ligands that appear to bind IDPs with relatively weak mM affinities as assessed by residue-level NMR chemical shift perturbations appear to bind substantially tighter, in the low μM affinity range, using other spectroscopic techniques such as SPR and bilayer interferometry^{27-28,30}. Understanding these relationships well enough to quantitatively compare simulated and spectroscopically measured binding affinities will likely require extensive experimental and computational benchmarking. Simulated K_D values are therefore most meaningfully compared to simulated K_D values calculated with the same distance thresholds and may not be directly comparable to experimental K_D values measured from biophysical experiments and spectroscopic techniques beyond ranking the relative affinities of ligands in a series.

We and others³² have found that the distance threshold used to define intermolecular contacts has little effect on the ratios of K_D values and contact probabilities calculated for a series of small molecule ligands in separate binding simulations. The distance threshold employed in the definition of bound conformations can therefore be thought of as a scaling factor for simulated K_D values and the optimal distance threshold for comparisons to experimental K_D values is likely to vary based on the experimental techniques used. In single replica unbiased MD simulations, one can utilize the residence time of ligand binding events to differentiate transient encounters from more stable binding events³², but applying residence time based thresholds for bound conformations is less straightforward when using enhanced sampling techniques such as REST2⁵³. In a previous study of small molecules binding a 20-residue fragment of α -synuclein that utilized unbiased MD simulations and the same force fields employed in this investigation³², a broad distribution of residence times, from 1ns-2 μs was observed. Based on the higher simulated binding affinities of the compounds studied here, we speculate that the distribution of residence times observed in unbiased MD simulations of EPI-002 and EPI-7170 binding Tau-5R₂_R₃ would be shifted to longer residence times.”

4. In figure 2C, should the diagonal not show the same data as Fig. 2A? The maximum of the color bar in 2C is at 0.25, but the data in 2A go up to 0.4. It's possible that 2C is referring only to the "bound ensemble" as labelled and not the whole simulation, but then

one would expect the contact probabilities to be higher and not lower.

In Figure 2C, the diagonal does show the same data as Figure 2A. We have chosen a maximum value of 0.25 to visualize the data in 2C even though the values on diagonal are higher, so small conditional contact probabilities are more clearly resolved off the diagonal.

Reviewer #4 (Remarks to the Author):

This is a manuscript that uses all-atom molecular dynamics computer simulations to elucidate atomically detailed binding mechanisms of EPI-002 and a more potent analogue, EPI-7170, to the known binding site of EPI-002 in Tau-5 of the amino-terminal domain (NTD) of the androgen receptor (AR) that was revealed previously by NMR. Two of the three regions (R2 and R3) of AR-NTD known to interact with EPI-002 are utilized while omitting R1, in spite of published data that all three regions (residues 354-448) must simultaneously be present for EPI-002 to bind (de Mol et al 2016). EPI-7170 has been shown by others to be more potent than EPI-002 in biological assays for blocking AR transcriptional activity, but it has not been shown that EPI-7170 binds to AR. EPI-7170 differs from EPI-002 by having two chlorine atoms on one of the phenol rings and in place of an alcohol has a MeO₂SHN group. These studies here are restricted to all-atom explicit solvent MD simulation of Tau-5R2-R3 and support published biological data in terms of EPI-7170 being more potent than EPI-002. The authors suggest specific amino acid residues within R2 and R3 of the AR-NTD and mechanisms that may underlie this difference in potency. No biological data is provided to support conclusions of the importance of these specific residues identified in the potential binding mechanisms of EPI-7170 to AR-NTD.

Concerns

1) Biological support of the conclusion that the aromatic residues W397 and W433 are important in EPI-7170 binding to the AR are lacking, or that they impact potency. It is not been proven that EPI-7170 binds AR.

We thank the reviewer for raising two important points. The first is regarding the biological activity of mutants predicted to have a lower affinity to EPI-002 and EPI-7170. We emphasize that the current manuscript is a computational study, with the goal of providing atomistic binding mechanisms of EPI-002 and EPI-7170, consistent with previous NMR and in vitro cellular results, that will inform the design of future computational, in vitro and in vivo experiments. Work is currently ongoing in the Robustelli and Salvatella laboratories to design and computationally and experimentally characterize AR mutants and new small molecules ligands with increased and decreased affinities for EPI ligands based on the simulations and ensembles reported here, including mutations to residues W397 and W433 in various contexts. These follow-up studies

represent a substantial on-going research effort, and we believe such experimental are outside the scope of this investigation.

The reviewer's second point is that it has not been proven that EPI-7170 binds AR. The Salvatella laboratory has indeed attempted to demonstrate this interaction with NMR spectroscopy, however, as noted in previous investigations, EPI-7170 was found to be too insoluble to enable such measurements. However, given the large amount of biophysical, in vitro, and in vivo data on small molecules with the same bisphenol A scaffold as EPI-7170 and EPI-002, we believe that it is highly likely that EPI-7170 binds to the Tau-5 region of AR in a similar fashion to the other compounds in the EPI family. Importantly, NMR spectroscopy was recently used to show the direct interaction between the clinical candidate EPI-7386 (which has an undisclosed chemical structure, but was generated in the same ligand series as EPI-7170) with the sidechains of residues W397 and W433 in Tau-5. (Reference 46 in the manuscript: Hong, N.H., Le Moigne, R., Pearson, P., Lauriault, V., Banuelos, C.A., Mawji, N.R., Tam, T., Wang, J., Virsik, P., Andersen, R.J. and Sadar, M.D., 2020. The preclinical characterization of the N-terminal domain androgen receptor inhibitor, EPI-7386, for the treatment of prostate cancer. European Journal of Cancer, 138, p.S51.)”

As mentioned in our response to a related point by Reviewer 2 we have updated the discussion of the manuscript on page 24-25 to reflect these points:

“NMR measurements of the interaction between EPI-7170 and Tau-5 have yet-to-be reported, which may be the result of the relative insolubility of EPI-7170 relative to EPI-002⁴². A number of small molecule ligands that bind IDPs have been found to be relatively insoluble at concentrations required for biophysical experiments, in particular NMR spectroscopy³². Obtaining sufficiently soluble small molecules drugs is often a challenge in drug discovery campaigns targeting folded proteins, where active ligands can have very tight picomolar-nanomolar affinities. As many small molecule disordered proteins ligands that have biological activity have been found to have relatively low mM-mM affinities in biophysical assays, obtaining sufficient solubility for detailed biophysical characterization may be particularly challenging in drug discovery campaigns for disordered targets. This suggests that solubility may be a ubiquitous problem when studying small molecules that bind IDPs and underscores the value of combining insights from experimental and computational studies when studying compounds near the detectable solubility limits of biophysical experiments. In this investigation, we have accessed the accuracy our simulations of apo Tau-5_{R2_R3} and EPI-002 Tau-5_{R2_R3} binding by comparisons with previously measured NMR data. The consistency between our simulations and NMR data suggests that the simulation methods and force fields utilized are well suited to describe the conformational properties of Tau-5_{R2_R3} and the interactions between Tau-5_{R2_R3} and EPI compounds with a common bisphenol-A scaffold. Given the relatively small differences in the chemical structures of EPI-002 and EPI-7170 and quality of ligand molecular mechanics force fields^{32,63}, we expect that our simulation model will be capable of discerning differences in their binding modes as observed in a previous study³².”

2) These studies would benefit from inclusion of the more structurally related compounds such as BADGE-2H2O and iodo-EPI-002. BADGE-2H2O has no biological activity (Andersen et al 2010) and is only one atom different from EPI-002; whereas iodo-EPI-002

compound has been proven to bind to the AR-NTD, is extremely potent, and also is only one atom different from EPI-002 (Imamura et al 2016). The EPI-7170 compound has not been proven to bind to the AR and has a substantially differing chemical structure from EPI-002.

We thank the reviewer for this excellent suggestion. We have now run long timescale REST2 MD simulations of BADGE-2H₂O and Iodo-EPI-002 interacting with Tau-5_{R2_R3} and have included the results in the revised manuscript in the new results section “Binding Mode Comparisons of EPI-002 and EPI-7170 with Iodo-EPI-002 and BADGE:2H₂O”. Further, we have also more clearly articulated our hypothesis that the non-covalent binding affinity of EPI ligands directs site-specific covalent attachment to specific cysteines in Tau-5, and that tighter non-covalent binding affinities of EPI ligands will increase the covalent reactivity of these ligands at specific cysteines. We have presented additional analyses and discuss this hypothesis in the new results section “Non-covalent binding of EPI-7170 & EPI-002 increases the proximity of their weakly reactive chlorohydrin groups to the nucleophilic thiol of CYS404 in collapsed helical states of Tau-5_{R2_R3}” and discussed the results of our BADGE:2H₂O binding simulations in terms of this hypothesis. The text of these sections is included below. Additionally, we have added 4 new figures to the Supplementary information (Fig S31-S34) and an additional table (Table S1) to describe the results of these simulations.

“Non-covalent binding of EPI-7170 & EPI-002 increases the proximity of their weakly reactive chlorohydrin groups to the nucleophilic thiol of CYS404 in collapsed helical states of Tau-5_{R2_R3}”

EPI-002 and EPI-7170 both possess a chlorohydrin group (Fig. 2). The chlorohydrin group of EPI-002 has been shown to be weakly reactive with cysteine residues in the AR-NTD, and it is hypothesized that covalent attachment of EPI compounds to Tau-5 may be required for its biological activity^{14,37,38}. It has previously been proposed that fast reversible non-covalent interactions of EPI compounds to different regions of the AR-NTD may drive covalent attachment to specific cysteines in AR³⁸. Based on previously reported NMR experiments on the AR-NTD that showed the largest backbone amide NMR CSPs in the R2 and R3 regions²⁴ in EPI-002 titrations, we hypothesize that non-covalent binding of EPI ligands may direct covalent attachment of these ligands to CYS404. CYS404 is found in the middle or the transiently helical R2 region (³⁹⁷WAAAAAQCRYG⁴⁰⁷) before the helix breaking GLY407 residue. If covalent reactivity at CYS404 is important for the biological activity of EPI ligands, we suspect that EPI ligands with non-covalent binding modes that more effectively localize their weakly reactive chlorohydrin groups to the nucleophilic thiol of CYS404 will be more potent AR inhibitors in vivo. We compare the distance distributions between the chlorohydrin chlorine atoms of EPI-7170 and EPI-002 to the sulfur atom of CYS404 of Tau-5_{R2_R3} in our ligand binding simulations in Fig. S31. We find that the higher non-covalent affinity of EPI-7170 relative to EPI-002 dramatically increases the proximity of its chlorohydrin group to CYS404 relative the distance distribution observed between the chlorohydrin group of EPI-002 and CYS404. We find that the chlorohydrin chlorine atom of EPI-7170 is within 10.0Å of the CYS404 thiol sulfur atom in 23.0% of simulation frames while the chlorine atom of EPI-002 is within 10.0Å of the CYS404 thiol sulfur atom in 9.6% of simulation frames. We note that in addition to increasing the proximity of the EPI ligand

chlorohydrin groups to CYS404, the compact nature of the ligand bound ensembles may further facilitate covalent attachment by sequestering the reaction from solvent water molecules. The conformational properties of covalent adducts of EPI compounds bound to Tau-5 will be the subject of future investigations.

Binding Mode Comparisons of EPI-002 and EPI-7170 with Iodo-EPI-002 and BADGE:2H₂O

To obtain additional insight into the inhibition mechanisms of EPI-002 and EPI-7170 we also conducted REST2 MD simulations of Tau-5_{R2_R3} in the presence of two additional ligands: Bisphenol A Diglycidic Ether:2H₂O (BADGE:2H₂O), and Iodo-EPI-002 (Figure S32). BADGE:2H₂O has an identical structure to EPI-002 aside from the replacement of the EPI-002 chlorohydrin group with a diol. This substitution eliminates the covalent reactivity of the compound, and cellular assays have shown that BADGE:2H₂O does not inhibit AR transcriptional activity^{37,38}. Iodo-EPI-002 differs from EPI-002 only by the addition of a single iodine atom on the bisphenol A phenyl ring closest to the chlorohydrin group and has been shown to be an ~10x more potent AR inhibitor in cellular AR transcriptional activity inhibition assays³⁹. A REST2 simulation of BADGE:2H₂O and Tau-5_{R2_R3} was run for 4.2 μ s/replica, for an aggregate simulation time of 56 μ s, and a REST2 simulation of Iodo-EPI-002 and Tau-5_{R2_R3} was run for 5.0 μ s/replica, for an aggregate simulation time of 80 μ s. The simulated properties of Tau-5_{R2_R3} observed in these simulations are compared to the apo Tau-5_{R2_R3} simulations in Table S1 and Fig. S32-S33. Simulation and convergence analyses for these simulations are reported in Fig S32-S33.

We observe that BADGE:2H₂O has a simulated K_D of 5.61 \pm 0.29 mM, which is slightly larger than, but within statistical uncertainty estimates of, the simulated K_D values of EPI-002 (5.24 \pm 0.43 mM). We observe that most simulated properties are within statistical uncertainty estimates of the simulated properties of EPI-002 (Table S1, Figure S32-S33) and that the intermolecular interactions between Tau-5_{R2_R3} and each ligand observed in the bound ensembles are extremely similar (Figure S34). The similar simulated binding affinities and properties of the bound ensembles of EPI-002 and BADGE:2H₂O are consistent with the hypothesis that differences in their biological potency may strictly result from a lack of covalent reactivity, rather than from differences in their non-covalent binding affinity or non-covalent binding mechanisms. These results are consistent with the hypothesis that covalent attachment is essential for the biological activity of EPI AR-NTD inhibitors.

Iodo-EPI-002 has been shown to be substantially more potent than EPI-002 in cellular AR inhibition assays³⁹. Iodo-EPI-002 was shown to have IC₅₀ values of ~1 μ M in prostate-specific antigen luciferase reporter assays for inhibition of endogenous AR transcriptional activity³⁹ compared to IC₅₀ values of ~10 μ M for EPI-002 in similar assays³⁷⁻³⁸. EPI-7170 was also found to have an IC₅₀ value of ~1 μ M in cellular prostate-specific antigen luciferase reporter assays⁴⁶. We found that the simulated K_D of Iodo-EPI-002 (1.89 \pm 0.13mM) is extremely similar to the simulated K_D of EPI-7170 (1.92 \pm 0.15mM). Interestingly, we observe that the Iodo-EPI-002:Tau-5_{R2_R3} bound ensemble has a larger fraction helix (38.4 \pm 1.7%) and helical globule population (79.2 \pm 3.3%) compared the EPI-7170:Tau-5_{R2_R3} bound ensemble (fraction helix of 32.8 \pm 0.5% and helical globule population of 61.1 \pm 4.5%). We observe that the populations of intermolecular

interactions in the ligand Iodo-EPI-002:Tau-5_{R2_R3} bound ensembles are very similar to those observed in EPI-7170:Tau-5_{R2_R3} bound ensemble (Figure S34). These simulation results suggest that Iodo-EPI-002 and EPI-7170 bind Tau-5 with similar binding mechanisms and similar affinities and are consistent with the similar potencies observed in cellular AR transcriptional inhibition assays.

We also have added the following to the discussion session on page 25 of the revised manuscript

“The chlorohydrin group of EPI-002 has been found to be weakly covalently reactive with Tau-5, and it is hypothesized that covalent attachment of EPI compounds to Tau-5 is required for their biological activity^{14,37-39}. We have simulated Tau-5_{R2_R3} binding the compound BADGE:2H₂O, which replaces the chlorohydrin group EPI-002 with an unreactive diol. We observe that EPI-002 and BADGE:2H₂O have very similar simulated binding affinities, and that the EPI-002:Tau-5_{R2_R3} and BADGE:2H₂O:Tau-5_{R2_R3} bound ensembles are extremely similar, in agreement with the hypothesis that covalent reactivity is essential for AR-NTD inhibition in cellular contexts. In our simulations, we observe that reversible non-covalent binding of EPI-002 and EPI-7170 at the dynamic interface between the R2 and R3 regions of Tau-5 increases the proximity of their weakly reactive chlorohydrin groups to the thiol group of cysteine 404, and that the higher affinity binding of EPI-7170 more effectively localizes its chlorohydrin group to the cysteine 404 thiol. We hypothesize that the non-covalent affinity of EPI compounds to this region may drive preferential covalent attachment to this cysteine residue relative to other cysteine residues in the AR-NTD, which may further stabilize compact helical conformations of Tau-5 and inhibit the⁴³³WHTLF⁴³⁷ segment of the R3 helix from binding the RAP74 domain of the general transcription regulator TFIIF^{47,48,51,52}. In this scenario, we expect that compounds with higher non-covalent binding affinity to Tau-5_{R2_R3}, such as EPI-7170 and Iodo-EPI-002, will be more covalently reactive in cellular contexts. The conformational properties of covalent adducts of EPI compounds bound to Tau-5 will be the subject of future investigations.”

3) Restriction of the studies to only Tau-5R2-R3 raises uncertainty about whether similar results would be obtained with these compounds with an unrelated protein sequence. Previous NMR work published with EPI-002 showed less binding to Tau-1 containing more structured regions. As is there is no control for the protein sequence in these studies.

The reviewer raises an important outstanding question in disordered protein molecular recognition. Identifying the minimal sequence motifs that can confer affinity and specificity for ligand binding, and differentiating ligand features that confer specific vs. non-specific binding, is currently a large outstanding challenge in the field of disordered protein molecular recognition. The Robustelli laboratory is currently engaged in preliminary research efforts to provide insights into these questions,

We believe however, that in order to obtain statistically significant insights into this important question, one needs to simulate and experimentally test ligand binding to a large number of sequences, both randomly scrambled and rationally perturbed, and that this substantial research effort is outside the intended scope of the present study.

4) AR-NTD is rich in cysteine residues which play an important role in protein structure through disulfide bridges. Some discussion about this in relation to the presented work would be helpful.

5) The binding site contains residues that are known to be post-translationally modified (for example pS424). Some discussion on such modifications and limitations of the modeling approach would be beneficial.

We have added additional discussion of these points, and added additional references to the manuscript, in the discussion section on Page 28:

“Finally, we note that the three cysteine residues in the Tau-5 region of AR and four additional cysteine residues in the AR-NTD may also form disulfide bonds in certain cellular conditions. Additionally, several post-translation modifications which substantially alter the cellular activity AR, such as phosphorylation at Serine 424 in the R3 helix have been discovered^{52,70-72}. The conformational ensembles presented here, along with future computational and experimental studies of the conformational ensembles of post-translationally modified Tau-5 constructs and Tau-5 covalent adducts, may be helpful for understanding the impact of these modifications on AR function *in vivo*.”

Additional References Added:

70. Gioeli, D., Ficarro, S.B., Kwiek, J.J., Aaronson, D., Hancock, M., Catling, A.D., White, F.M., Christian, R.E., Settlage, R.E., Shabanowitz, J. and Hunt, D.F., 2002. Androgen receptor phosphorylation: regulation and identification of the phosphorylation sites. *Journal of Biological Chemistry*, 277(32), pp.29304-29315.

71. Koryakina, Y., Ta, H.Q. and Gioeli, D., 2014. Androgen receptor phosphorylation: biological context and functional consequences. *Endocrine-related cancer*, 21(4), pp.T131-T145.

72. Liao, M., Zhou, Z.X. and Wilson, E.M., 1999. Redox-dependent DNA binding of the purified androgen receptor: evidence for disulfide-linked androgen receptor dimers. *Biochemistry*, 38(30), pp.9718-9727.

6) References about covalent binding need to be corrected in the last paragraph of the Discussion that incorrectly cite ref 37, whereas references 38 and 39 address this point.

*We have corrected these references. We note that we have included reference 37 (Andersen, R.J., Mawji, N.R., Wang, J., Wang, G., Haile, S., Myung, J.K., Watt, K., Tam, T., Yang, Y.C., Bañuelos, C.A. and Williams, D.E., 2010. Regression of castrate-recurrent prostate cancer by a small-molecule inhibitor of the amino-terminus domain of the androgen receptor. *Cancer cell*, 17(6), pp.535-546) in instances where we discuss the activity of BADGE:2H₂O, as this work includes results for this compound (referred to as compound 185-9-1 in this manuscript).*

REVIEWERS' COMMENTS

Reviewer #3 (Remarks to the Author):

The authors have adequately addressed the points raised.

We thank the reviewers for their comments. We agree that additional experimental validation of the binding modes observed in our simulations would be very valuable, unfortunately, the insolubility of the compound EPI-7170 precludes the possibility of the additional requested biophysical measurements requested by Reviewer 1 at this time. We intend to use similar methods to those requested to further experimentally characterize the binding mechanisms of more soluble androgen receptor ligands in future work. We have explained this below and updated our final manuscript to reflect this.

Reviewer #1 (Remarks to the Author)

The authors have addressed most of the reviewers' concerns. However, there is still no experimental evidence for EPI-7170's direct binding with AR transactivation domain. Although more compounds were simulated, they are structurally different from EPI-7170 and could not be extended to and used to confirm that EPI-7170 binds AR. If the solubility of EPI-7170 is not enough for NMR study, other direct binding assays should be performed. To confirm the collapsed structure after compound binding, other experimental assays should be used, such as SAXS, dynamic light scattering, etc.

As noted by the reviewer, The Salvatella laboratory has attempted to demonstrate direct interaction between EPI-7170 and AR with NMR spectroscopy, but EPI-7170 was found to be too insoluble to enable such measure. Unfortunately, the insolubility of EPI-7170 compound also makes experiments such as SAXS and dynamic light scattering extremely difficult to perform on this system.

We agree that additional biophysical characterization of AR binding of ligands similar to those studied in this work is of great interest but we believe that this substantial experimental undertaking is outside the scope of the current manuscript. Additional experimental biophysical characterization using NMR, SAXS, and dynamic light scattering of more soluble AR ligands will be the subject of future investigations.

We have updated the text of the revised manuscript to reflect this:

“We hypothesize that as EPI-7170 shares a common Bisphenol A scaffold with a large number of EPI derivatives that have been shown to have in vitro and in vivo interactions with AR and EPI-7170 has been shown to have a similar biological activity to these ligands^{37-39,46} it is very likely that EPI-7170 also directly binds the Tau-5 region of AR. *The relative insolubility of EPI-7170 may make biophysical measurements to confirm direct AR binding difficult to perform and interpret, but further biophysical characterization of the binding modes of more soluble ligands that bind the Tau-5 region of AR are of great interest, and will be subject of future investigations.*”

Reviewer #3 (Remarks to the Author)

The authors have adequately addressed the points raised.

We thank the reviewer for their comments and for rereviewing the manuscript